# LTP induction by structural rather than enzymatic functions of CaMKII

Jonathan E. Tullis[1,4], Matthew E. Larsen[1,2,4], Nicole L. Rumian[1,2], Ronald K. Freund[1], Emma E. Boxer[1,2], Carolyn Nicole Brown[1], Steven J. Coultrap[1], Howard Schulman[3], Jason Aoto[1,2], Mark L. Dell'Acqua[1,2] & K. Ulrich Bayer[1,2✉]

Learning and memory are thought to require hippocampal long-term potentiation (LTP), and one of the few central dogmas of molecular neuroscience that has stood undisputed for more than three decades is that LTP induction requires enzymatic activity of the $Ca^{2+}$/calmodulin-dependent protein kinase II (CaMKII)[1–3]. However, as we delineate here, the experimental evidence is surprisingly far from conclusive. All previous interventions inhibiting enzymatic CaMKII activity and LTP[4–8] also interfere with structural CaMKII roles, in particular binding to the NMDA-type glutamate receptor subunit GluN2B[9–14]. Thus, we here characterized and utilized complementary sets of new opto-/pharmaco-genetic tools to distinguish between enzymatic and structural CaMKII functions. Several independent lines of evidence demonstrated LTP induction by a structural function of CaMKII rather than by its enzymatic activity. The sole contribution of kinase activity was autoregulation of this structural role via T286 autophosphorylation, which explains why this distinction has been elusive for decades. Directly initiating the structural function in a manner that circumvented this T286 role was sufficient to elicit robust LTP, even when enzymatic CaMKII activity was blocked.

Long-term potentiation (LTP) is impaired by pharmacological calmodulin-dependent protein kinase II (CaMKII) inhibition with two different classes of mechanistically distinct inhibitors[4,5], by knockout of the CaMKIIα isoform that is predominant in brain[6], by a mutation that prevents ATP binding[7] or by T286A mutation that prevents the T286 autophosphorylation (pT286)[8] that generates $Ca^{2+}$-independent 'autonomous' CaMKII kinase activity[15,16]. However, all these LTP-inhibiting interventions with enzymatic CaMKII activity also interfere with CaMKII binding to the NMDA-type glutamate receptor subunit GluN2B. (1) Calmodulin-competitive inhibitors, such as KN93 or KN62, prevent the stimulus that induces both activity and GluN2B binding[9,10]; (2) the peptide inhibitors tatCN21, tatCN19o, AIP and AC3-I bind to the CaMKII T site, which is also the binding site for GluN2B[9,11,12]; (3) the K42M and K42R mutants prevent nucleotide binding to CaMKII, which is a requirement not only for activity but also for effective GluN2B binding[13,14]; and (4) whereas the T286A mutation does not completely block CaMKII binding to GluN2B, it significantly impairs the stimulation-induced binding of CaMKII to GluN2B within cells and the resulting synaptic CaMKII accumulation in neurons[13,17]. Thus, we set out to determine the relative contributions of CaMKII enzymatic activity versus GluN2B binding in LTP induction.

## Light-induced paCaMKII binding to GluN2B

Direct photoactivation of paCaMKII is sufficient to induce structural long-term potentiation (sLTP) in hippocampal neurons even in the absence of neuronal stimulation[18], apparently indicating that CaMKII enzymatic activity would be sufficient for LTP induction. However, similar to stimulation by $Ca^{2+}$/CaM, photoactivation of paCaMKII involves displacement of the autoinhibitory domain to expose not only the substrate binding surface (S site)[19] but also the neighbouring GluN2B-binding surface (T site)[9]. (Note that these two sites form a continuous groove and can act together in protein binding[11,20], with GluN2B binding thought to progress from an initial S-site binding[11] to a more stable T-site binding mode[9,11]. The CaMKII binding site on GluN2B around S1303 shares homology with the CaMKII regulatory domain region around T286, and the T-site binding mode is exemplified by the T286 interaction with the kinase domain in the basal state, whereas the S-site binding mode is exemplified by the T286 interaction during its autophosphorylation by a neighbouring kinase subunit within the CaMKII holoenzyme.) Thus, photoactivation of paCaMKII may additionally directly promote binding to GluN2B (Fig. 1a). To test this, we co-expressed paCaMKII labelled with green fluorescent protein (GFP) and a membrane-targeted mCh-GluN2B-c tail in HEK293 cells and monitored colocalization before and after blue-light stimulation. We observed robust colocalization of GFP–paCaMKII and mCh-GluN2B-c tail but only after photoactivation (Fig. 1b,c and Extended Data Fig. 1a), indicating that photoactivation of paCaMKII indeed also promotes GluN2B binding. This light-induced binding of paCaMKII to the GluN2B-c tail was also corroborated by our in vitro binding assay (Extended Data Fig. 1b,c). To further validate this, we utilized two mutations that render CaMKII unable to bind GluN2B: mutation of the CaMKII T site (I205K)[9] or the CaMKII ATP-binding site (K42M)[13]. Neither of these mutant paCaMKII constructs showed

[1]Department of Pharmacology, University of Colorado Anschutz Medical Campus, Aurora, CO, USA. [2]Program in Neuroscience, University of Colorado Anschutz Medical Campus, Aurora, CO, USA. [3]Department of Neurobiology, Stanford University School of Medicine, Stanford, CA, USA. [4]These authors contributed equally: Jonathan E. Tullis, Matthew E. Larsen. ✉e-mail: ulli.bayer@cuanschutz.edu

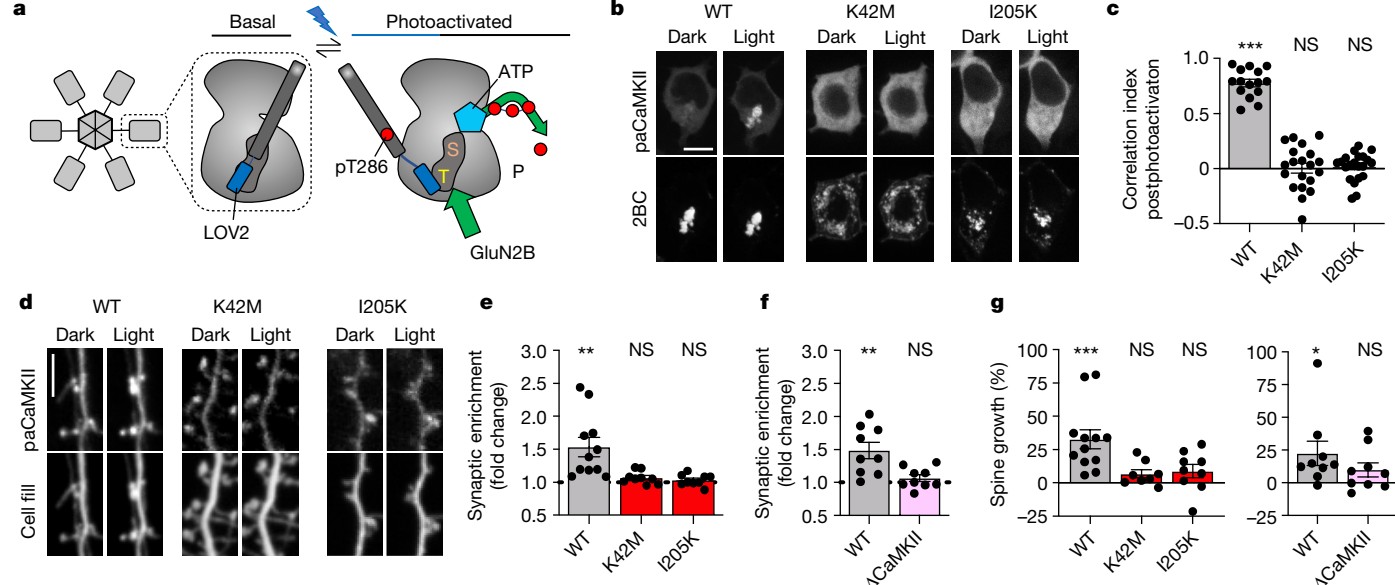

**Fig. 1 | paCaMKII photoactivation-induced sLTP requires GluN2B binding.**
Data are presented as mean values ± s.e.m. **a**, Schematic of how photoactivation of paCaMKII allows access to both S and T sites, thereby enabling both enzymatic CaMKII activity and GluN2B binding. **b**, Representative confocal microscopy images of HEK293 cells co-expressing GFP–paCaMKII (WT, K42M or I205K) and pDisplay-mCherry-GluN2B-c tail (mCh-2BC) before and after photoactivation. Scale bar, 10 μm. Quantification is shown in panel **c**. **c**, Pearson's correlation (correlation index) of GFP–paCaMKII WT or mutants and mCh-2BC 5 min after photoactivation (*n* = 15, 20, 21 cells; one-sample *t*-test). WT *P* < 0.0001; K42M *P* = 0.9468; I205K *P* = 0.5885. ***P* < 0.001. **d**, Representative confocal microscopy images of dissociated rat hippocampal cultures expressing GFP–paCaMKII (WT, K42M or I205K) and mCherry cell fill before and after photoactivation.

Scale bar, 5 μm. Quantification is shown in panel **e**. **e**, Fold change of paCaMKII WT and mutant synaptic enrichment values 15 min after photoactivation (*n* = 11, 9, 9 cells; one-sample *t*-test). WT *P* = 0.0042; K42M *P* = 0.0568; I205K *P* = 0.3523. ***P* < 0.01. **f**, Fold change of paCaMKII synaptic enrichment values 15 min after photoactivation in WT and GluN2B^ΔCaMKII mice (*n* = 9, 9 cells; one-sample *t*-test). WT *P* = 0.0038; GluN2B^ΔCaMKII *P* = 0.2172. ***P* < 0.01. **g**, Changes in the dendritic spine area were measured after paCaMKII photoactivation via mCherry cell fill fluorescence intensity within a spine (*n* = 12, 8, 9, 9, 9 cells; one-sample *t*-test). In the left panel, WT *P* = 0.008; K42M *P* = 0.0692; and I205K *P* = 0.1236. In the right panel, WT *P* = 0.0481 and GluN2B^ΔCaMKII *P* = 0.1406. **P* < 0.05, ****P* < 0.001. NS, not significant.

colocalization with mCh-GluN2B after photoactivation in HEK cells (Fig. 1c and Extended Data Fig. 1d). In rat hippocampal neurons, regulated GluN2B binding is required for normal LTP and is responsible for LTP stimuli-inducing movement of CaMKII to dendritic spines, the post-synaptic compartments of excitatory synapses[9,21–23]. Photoactivation of GFP–paCaMKII directly induced its movement to synapses, even in the absence of any other stimulation (Fig. 1d,e). This synaptic movement of GFP–paCaMKII was completely abolished by two distinct CaMKII mutations, I205K and K42M (Fig. 1d,e), both of which inhibit CaMKII binding to GluN2B[9,13]. Similarly, in mouse hippocampal neurons, the synaptic movement induced by photoactivation of GFP–paCaMKII was seen only in cultures from wild-type (WT) mice but not in cultures from mice with the GluN2B^ΔCaMKII mutation that prevents CaMKII binding[22,24] (Fig. 1f and Extended Data Fig. 1e). These results indicate that photoactivation causes the same type of synaptic enrichment of CaMKII as LTP stimuli.

## sLTP by paCaMKII requires GluN2B binding

In parallel, we tested whether spine growth after photoactivation of paCaMKII[18] required GluN2B binding. Spine growth is considered a readout for sLTP and was here monitored in live neurons using mCherry as cell fill. Photoactivation of GFP–paCaMKII WT induced robust spine growth in rat hippocampal neurons, and this effect was blocked by the I205K and K42M mutations that inhibit GluN2B binding (Fig. 1d,g). Whereas the K42M mutation additionally abolishes enzymatic CaMKII activity, the I205K mutation does not[19], indicating that a structural function such as GluN2B binding is required for photoactivation-induced spine growth and that enzymatic activity of paCaMKII alone is not sufficient. Indeed, photoactivation-induced spine growth was also seen

in hippocampal neurons from WT mice but not in cultures from mice with the GluN2B^ΔCaMKII mutation that prevents CaMKII binding (Fig. 1g and Extended Data Fig. 1e).

## AS283 does not block GluN2B binding

Normal GluN2B binding by CaMKII requires stimulation by Ca^2+/CaM as well as occupation of its ATP-binding pocket[9,13]. However, this occupation can be fulfilled by ATP, ADP or ATP-competitive inhibitors, such as the broad-spectrum kinase inhibitors staurosporine and H7[13,25]. Thus, we tested if this is also the case for the new ATP-competitive and CaMKII-selective inhibitors AS283 (unpublished) and AS105 (ref. 26) (Fig. 2a and Extended Data Fig. 2a). Indeed, AS105 inhibits the enzymatic activity of CaMKII without inhibiting either GluN2B binding or stimulation-induced movement to excitatory synapses (Extended Data Fig. 2b–f); however, this inhibitor is no longer available in sufficient quantities. Thus, we additionally tested AS283, which has even further improved CaMKII selectivity compared with AS105 (Extended Data Table 1). AS283 effectively inhibited both phosphorylation of GluA1 S831 (pS831) and autophosphorylation of T286 (pT286) in biochemical assays in vitro, although the exogenous pS831 appeared to be inhibited slightly more potently than the pT286 that occurs within the holoenzyme, with apparent inhibitor constant ($K_i$) values of 17 and 71 nM, respectively (Fig. 2b). To calculate the $K_i$ values, a CaMKII Michaelis-Menten constant ($K_m$) for ATP of 33.3 μM was used (Extended Data Fig. 1c). Importantly and in contrast to tatCN21 (5 μM), AS283 (10 μM) did not inhibit CaMKII binding to GluN2B in our in vitro binding assays in the presence of ADP (Fig. 2c) or in our colocalization assays in HEK cells (Fig. 2d). In neurons, AS283 did not interfere with the GluN2B-dependent CaMKII

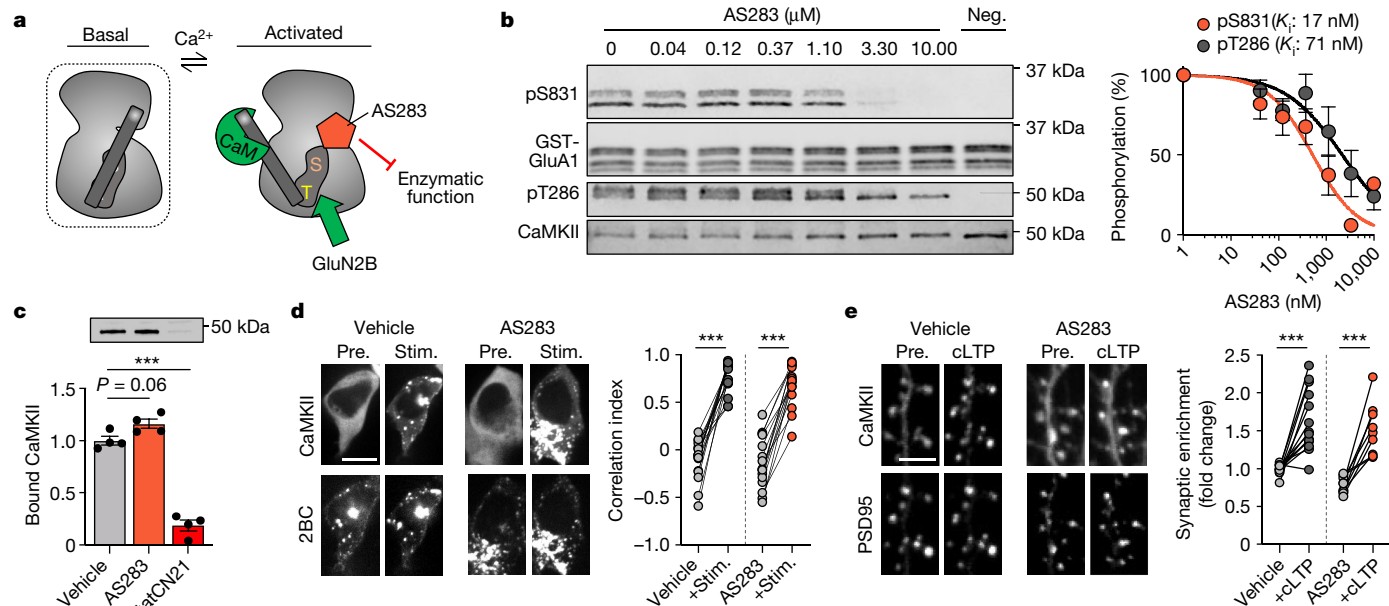

**Fig. 2 | AS283 inhibits CaMKII enzymatic function but does not impair GluN2B binding or synaptic localization.** Data are presented as mean values ± s.e.m. **a**, Schematic of the AS283 inhibitor mechanism of action: ATP-competitive inhibition blocks enzymatic activity while mimicking the nucleotide binding requirement to allow interaction with GluN2B. **b**, Representative immunoblot and quantification of in vitro kinase reactions at 30 °C in 1 mM ATP measuring purified CaMKIIα phosphorylation of GST–GluA1 S831 with varying concentrations of AS283 (0.41–10 μM; $n = 4$ independent concentration curves). For a negative control (Neg.), ATP was omitted. $K_i$ values were derived from half-maximal inhibitory concentration (IC50) values of AS283-mediated inhibition of CaMKII phosphorylation of S831 and T286 using an experimentally derived $K_m$ value of 33.3 μM (Extended Data Fig. 2b). **c**, Binding of purified CaMKIIα to immobilized GST–GluN2B-c was induced by Ca²⁺/CaM in the presence of 1 mM ADP alone or with 10 μM AS283 or 5 μM

tatCN21 ($n = 4$ independent samples; one-way analysis of variance (ANOVA) with Tukey's multiple comparisons test). ADP:AS283 $P = 0.0697$; ADP:tatCN21 $P < 0.0001$. ***$P < 0.001$. **d**, Confocal microscopy images of HEK293 cells co-expressing GFP–CaMKII and mCherry-2BC. Correlation indices before (Pre.) and after (Stim.) ionomycin-induced colocalization of GFP–CaMKII and mCherry-2BC under vehicle or 10 μM AS283 conditions ($n = 18, 16$ cells; repeated measures (RM) two-way ANOVA with Šídák's multiple comparisons test). Scale bar, 10 μm. Vehicle $P < 0.0001$; AS283 $P < 0.0001$. ***$P < 0.001$. **e**, Confocal microscopy images of dissociated rat hippocampal cultures expressing GFP–CaMKII and mCh-PSD95 intrabody in the presence of 10 μM AS283 before and after cLTP (100 μM glutamate, 10 μM glycine; 45 s). CaMKII synaptic enrichment was measured before and after cLTP ($n = 14,9$ cells; RM two-way ANOVA with Šídák's multiple comparisons test). Scale bar, 5 μm. Vehicle $P < 0.0001$; AS283 $P < 0.0001$. ***$P < 0.001$.

movement to excitatory synapses in response to chemical long-term potentiation (cLTP) stimuli (Fig. 2e). Thus, AS105 and AS283 are tools that inhibit enzymatic activity of CaMKII with high potency and selectivity without disrupting its binding to GluN2B. Therefore, they can be used to distinguish between enzymatic and structural functions of CaMKII.

## AS283 does not block LTP induction

With AS283 established as an effective CaMKII inhibitor that does not block binding to GluN2B, we decided to test its effect on LTP induction at the hippocampal CA3 to CA1 synapse in WT mice using field recordings in acute slices. Under our experimental conditions, LTP was effectively blocked by tatCN21 (5 μM) (Fig. 3a,b) as expected from our previous studies[5]. Previous studies have also indicated LTP inhibition with broad-spectrum ATP-competitive inhibitors, such as H7 (ref. 4). By contrast, incubation with the CaMKII-selective AS283 (10 or 30 μM) failed to block LTP induction (Fig. 3a,b and Extended Data Fig. 3a,b). This result was highly surprising because at least one enzymatic CaMKII reaction is thought to be required for LTP induction: autophosphorylation of CaMKII at T286 (ref. 8). This suggested three possibilities. (1) AS283 is not effective at inhibiting pT286, (2) AS283 is not effective in hippocampal slices or (3) AS283 somehow circumvents the requirement for pT286. Somewhat surprisingly, our following experiments provide strong and convincing support for the latter (that is, that AS283 effectively circumvents the requirement of pT286 in LTP at CA1 synapses).

## AS283 restores LTP in T286A mutant mice

The possibility that AS283 might not effectively block pT286 was suggested by prior precedent; despite tatCN21 being more potent than KN93 at inhibiting exogenous CaMKII substrates, it has been described to be less effective at inhibiting autophosphorylation of T286 (ref. 12). However, although AS283 also inhibited pT286 with slightly less potency than an exogenous substrate, this inhibition was still very potent and effective, at least in vitro (Fig. 2a) and in hippocampal cultures (Extended Data Fig. 3c). Thus, we tested other possibilities of AS283 action. If AS283 blocks pT286 but not LTP because it also circumvents the requirement of pT286 for LTP, then AS283 should not only fail to block LTP in WT mice (as shown in Fig. 3a,b) but also restore the capacity for LTP induction in T286A mutant mice. Even though this scenario sounds somewhat unlikely, this was exactly what was observed experimentally; the addition of AS283 (10 μM) enabled LTP in slices from the T286A mutant mice (Fig. 3c,d). This result directly demonstrates the effectiveness of AS283 in slices and strongly supports the notion that LTP induction requires structural CaMKII functions rather than its enzymatic activity toward exogenous substrates.

## AS283 enhances CaMKII binding to GluN2B

As one function of pT286 in LTP is enhancing GluN2B binding, we examined the possibility that AS283 may circumvent pT286 function by directly enhancing GluN2B binding even in absence of pT286.

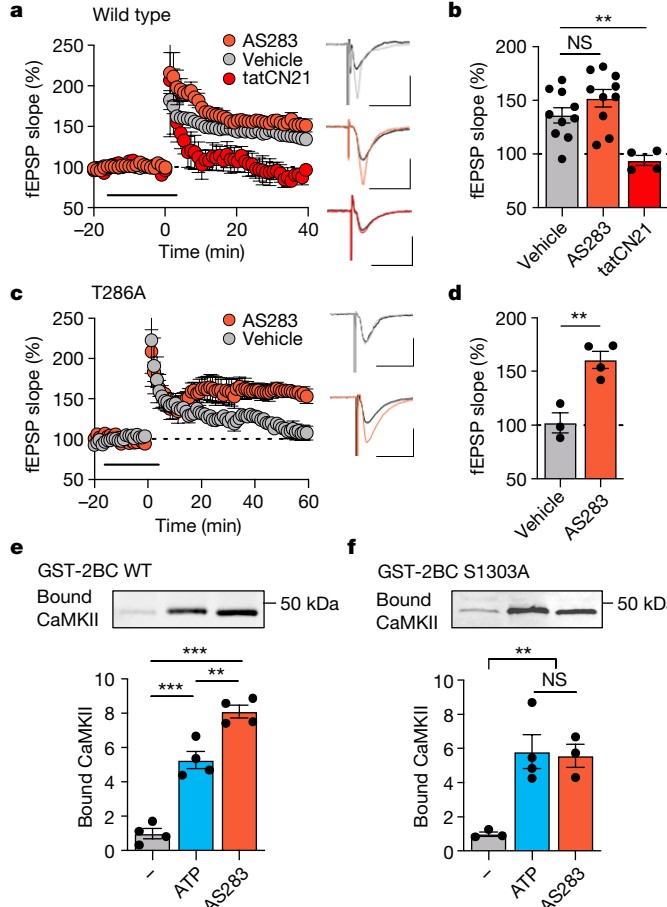

**Fig. 3 | AS283 does not inhibit LTP in WT mice and enables LTP in T286A mice.** Data are presented as mean values ± s.e.m. **a**, Two times HFS potentiates the CA3–CA1 Schaffer collateral pathway in WT mice. LTP was additionally measured after inhibition of CaMKII with 10–30 μM AS283 or 5 μM tatCN21 (incubated for 15 min before HFS and washed out 5 min after LTP induction; $n = 10, 10, 4$ animals). Scale bar, 0.5 mV by 20 ms. **b**, Quantification of LTP in WT mice after vehicle, AS283 or tatCN21 treatments ($n = 10, 10, 4$ animals; one-way ANOVA, Tukey's multiple comparisons test). Vehicle:AS283 $P = 0.1292$; vehicle:tatCN21 $P = 0.0097$; AS283:tatCN21 $P = 0.009$. $**P < 0.01$. **c**, LTP was measured by field excitatory post-synaptic potentials (fEPSP) in T286A mice after two times HFS under vehicle conditions and after inhibition of CaMKII by 10 μM AS283 during LTP induction ($n = 3, 4$ animals). Scale bar, 0.5 mV by 20 ms. **d**, Quantification of LTP in T286A mice after vehicle or AS283 treatment ($n = 3, 4$ animals; two-tailed Student's $t$-test). Vehicle:AS283 $P = 0.0048$. $**P < 0.01$. **e**,**f**, Binding of purified CaMKIIα to immobilized GST–GluN2B-c WT (**e**) or S1303A (**f**) was induced by $Ca^{2+}$/CaM without nucleotide, with 1 mM ATP or with 10 μM AS283. **e**, $n = 4, 4, 4$. **f**, $n = 3, 4, 3$. One-way ANOVA with Tukey's multiple comparisons test. **e**, −:ATP $P < 0.0001$; −:AS283 $P < 0.0001$; ATP:AS283 $P = 0.0018$. **f**, −:ATP $P = 0.0082$; −:AS283 $P = 0.0150$; ATP:AS283 $P = 0.9735$. $**P < 0.01$, $***P < 0.001$.

Indeed, when ATP was substituted with AS283 in our in vitro GluN2B binding assay, the $Ca^{2+}$/CaM-induced CaMKII binding was significantly enhanced, even though this prevents the occurrence of pT286 (Fig. 3e). Thus, even though both ATP and pT286 enhance CaMKII binding to GluN2B[13], AS283 can directly enhance this binding even more. As a consequence, in the presence of AS283, pT286 appears to be dispensable for proper GluN2B binding and LTP.

One possible mechanism by which AS283 could enhance GluN2B binding more than ATP is by suppressing CaMKII-mediated phosphorylation of GluN2B at S1303, as this phosphorylation within the CaMKII binding site on GluN2B reduces CaMKII binding[13,27]. Indeed,

when binding was tested using a phosphorylation-incompetent GluN2B S1303A mutant, the enhancing effects of AS283 and ATP on CaMKII binding were indistinguishable (Fig. 3f). Furthermore, for the WT GluN2B C terminus, AS283 indeed inhibited the CaMKII-mediated S1303 phosphorylation (Extended Data Fig. 3d), as expected.

## CaMKII F89G mutation as a powerful tool

AS283 is highly selective for CaMKII (Extended Data Table 1), and the pharmacogenetic restoration of the capacity for LTP induction in T286A mice by AS283 provided powerful and convincing support for our conclusion that structural functions rather than enzymatic functions of CaMKII are essential for LTP induction. Nonetheless, we sought further independent experimental support using a pharmacogenetic approach with the CaMKII F89G 'Shokat' mutation, which enlarges the ATP-binding pocket to enable selective ATP-competitive inhibition with NM-PP1, an inhibitor that does not affect WT kinases[28] (Fig. 4a). Surprisingly, the F89G mutation also directly reduced CaMKII enzymatic activity (Fig. 4b and Extended Data Fig. 4a); however, any remaining residual activity appeared to be still further reduced by 10 μM NM-PP1 (Fig. 4b), as expected. Similarly, although NM-PP1 did not affect pT286 for CaMKII WT, as expected, the F89G mutant did not show any significant pT286 at all (Extended Data Fig. 4a). In HEK cells, GFP–CaMKII F89G did not colocalize with mCh-GluN2B after ionomycin stimulation (Fig. 4c), similar to that seen for the kinase-dead K42M mutation that is impaired for binding of nucleotide (Extended Data Fig. 4b). However, the addition of 10 μM NM-PP1 fully enabled binding of CaMKII F89G to GluN2B in HEK cells (Fig. 4c). Similarly, NM-PP1 enabled binding of CaMKII F89G to GluN2B also in vitro (Extended Data Fig. 4c). Together, these results strongly indicate that the CaMKII F89G mutation dramatically reduces nucleotide binding (which inhibits both activity and GluN2B binding) but allows effective occupation of the mutated nucleotide binding pocket with NM-PP1 (which further reduces any residual enzymatic activity but now enables GluN2B binding). This conclusion also predicts that the LTP stimulus-induced movement of CaMKII to excitatory synapses should be (1) impaired in the CaMKII F89G mutant but (2) normalized by pharmacogenetic rescue with NM-PP1. Indeed, both predictions were observed experimentally (Fig. 4d,e). Whereas the F89G mutant impaired LTP-induced synaptic CaMKII localization to the same extent as the K42M mutant that also impairs ATP binding (Fig. 4d), this impaired localization was rescued by NM-PP1 only for the F89G but not the K42M mutant (Fig. 4e), demonstrating specificity and selectivity of the NM-PP1 effect.

## Restoring LTP in CaMKII F89G slices

To further validate our previous results demonstrating that only structural CaMKII functions are required for LTP induction, we decided to test pharmacogenetic rescue of LTP with NM-PP1 in hippocampal slices expressing the CaMKII F89G mutant. For this purpose, we re-expressed mScarlet-labelled CaMKII F89G mutant in the hippocampal CA1 region of CaMKIIα knockout mice. This was done by stereotactic injection of a packaged adeno-associated virus (AAV) construct, in which the expression of either mScarlet alone or the mScarlet-labelled mutant CaMKII is controlled by the CaMKIIα promoter (Fig. 5a). After 5 weeks of expression, acute slices were prepared and evaluated for proper localization of expression by imaging of mScarlet fluorescence (Fig. 5a and Extended Data Fig. 5a). In the CaMKIIα knockout slices that expressed the mScarlet control, LTP was significantly reduced (compare with Fig. 3) but not completely abolished (Fig. 5b,c), consistent with previous studies on these knockout mice[29,30]. As expected, NM-PP1 had no effect on the residual LTP in these control slices (Fig. 5b,c and Extended Data Fig. 5b). By contrast, in slices expressing the CaMKII F89G mutant, LTP

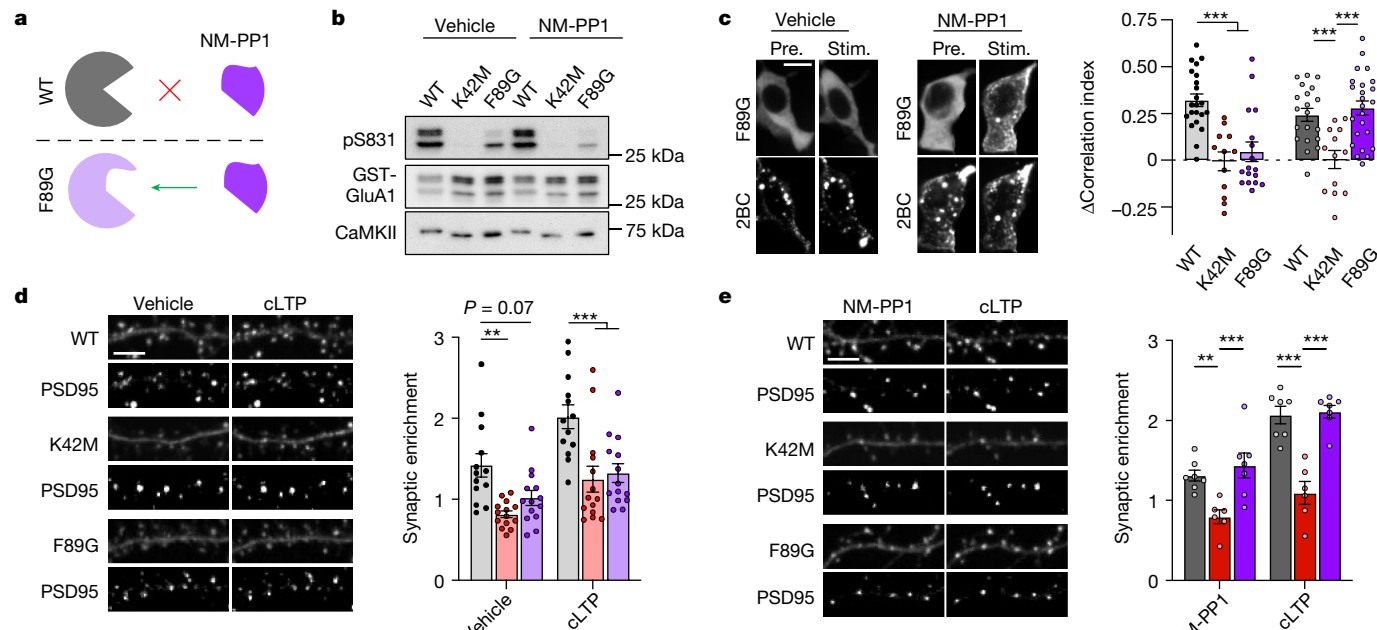

**Fig. 4 | The CaMKII mutation F89G impairs ATP binding but binds the ATP-competitive inhibitor NM-PP1 that restores GluN2B binding and LTP-induced synaptic translocation.** Data are presented as mean values ± s.e.m. **a**, Schematic of the pharmacogenetic method of ATP-competitive CaMKII inhibition; enlargement of the CaMKII ATP-binding pocket by the F89G mutation allows for selective binding of the ATP-competitive inhibitor NM-PP1. **b**, Immunoblots of in vitro kinase reactions at 30 °C including GFP–CaMKII WT, K42M and F89G with GST–GluA1-c tail. Reactions were performed either with vehicle control or with 10 μM NM-PP1 (n = 1) (related independent experiments are in Extended Data Fig. 4a). **c**, Confocal microscopy images of HEK293 cells co-expressing GFP–CaMKII F89G and mCherry-2BC in the presence of vehicle or 10 μM NM-PP1 before and after ionomycin stimulation. Correlation indices were measured before and after Ionomycin-induced colocalization of GFP–CaMKII (WT, K42M and F89G) with mCherry-2BC, represented as change in the correlation index (vehicle: n = 22, 17, 12 cells; NM-PP1: n = 20, 24, 13 cells; one-way ANOVA, Tukey's multiple comparisons test). Scale bar, 10 μm. For vehicle,

WT:F89G P < 0.0001, WT:K42M P < 0.0001 and F89G:K42M P = 0.7333. For NM-PP1, WT:F89G P = 0.7771, WT:K42M P = 0.0009 and F89G:K42M P < 0.0001. ***P < 0.001. **d**, Confocal microscopy images of dissociated rat hippocampal cultures expressing GFP–CaMKII (WT, K42M or F89G) and mCh-PSD95 intrabody before and after cLTP. CaMKII synaptic enrichment was measured before and after cLTP (n = 13, 14, 14 cells; RM two-way ANOVA with Šídák's multiple comparisons test). Scale bar, 5 μm. For basal, WT:K42M P = 0.0022, WT:F89G P = 0.0655 and K42M:F89G P = 0.5478. For cLTP, WT:K42M P < 0.0001, WT:F89G P = 0.0004 and K42M:F89G P = 0.9562. **P < 0.01, ***P < 0.001. **e**, Confocal microscopy images before and after cLTP as in panel **d** but in the presence of 10 μM NM-PP1. CaMKII synaptic enrichment was measured before and after cLTP (n = 7, 6, 7 cells; RM two-way ANOVA with Šídák's multiple comparisons test). Scale bar, 5 μm. For basal, WT:K42M P = 0.0079, WT:F89G P = 0.6909 and K42M:F89G P = 0.0009. For cLTP, WT:K42M P < 0.0001, WT:F89G P = 0.9582 and K42M:F89G P < 0.0001. **P < 0.01, ***P < 0.001.

was completely abolished (Fig. 5d,e), consistent with residual LTP being further suppressed by the dominant negative effect (on the remaining minor CaMKIIβ isoform in the CaMKIIα knockout)[30,31] that is exerted by a CaMKII mutant with impaired ATP binding[7] and thereby, is deficient in both activity and GluN2B binding. However, most importantly, in these slices, LTP induction was enabled by addition of the NM-PP1 inhibitor (10 μM) (Fig. 5d,e and Extended Data Fig. 5c–e) that further reduces enzymatic activity but enables GluN2B binding of the F89G mutant (Fig. 4). Thus, two independent lines of pharmacogenetic evidence topple current dogma by directly supporting the conclusion that LTP induction requires structural rather than enzymatic functions of CaMKII.

## GluN2B binding is sufficient for sLTP

Finally, we decided to utilize our two sets of pharmacogenetic tools in combination with the photoactivable paCaMKII to test whether structural CaMKII functions are not only the necessary but also the sufficient CaMKII functions in LTP. Specifically, we examined hippocampal neurons for sLTP (that is, the LTP-associated growth of dendritic spines) in response to direct photoactivation of paCaMKII without any other stimulation that could trigger additional signalling pathways. Similar to that seen with paCaMKII I205K or K42M (Fig. 1), photoactivation of paCaMKII T286A or F89G did not trigger any significant CaMKII

movement to spines or spine growth (Fig. 6). However, in the presence of the AS283 inhibitor (10 μM), photoactivation of the paCaMKII T286A mutant triggered robust CaMKII movement and spine growth to a similar extent as seen for paCaMKII WT (Fig. 6a–c). In a parallel finding, the addition of NM-PP1 (10 μM) enabled similar movement and spine growth after photoactivation of paCaMKII F89G (Fig. 6d–f). Thus, in both cases, suppressing enzymatic CaMKII activity while allowing or enhancing GluN2B binding enabled induction of sLTP. These results are consistent with the corresponding rescue of CA1 LTP in hippocampal slices (Figs. 3c,d and 5d,e). Additionally, as the photoactivation of paCaMKII does not trigger any other signalling pathways, these results indicate that the structural CaMKII functions are not only the necessary but also, the sufficient CaMKII functions for induction of sLTP.

## Discussion

Three independent lines of investigation showed that LTP induction requires structural CaMKII functions rather than enzymatic activity toward other substrate proteins (a schematic summary is in Extended Data Fig. 6); the direct photoactivation of CaMKII binding to GluN2B additionally indicated that these structural functions are not only the necessary but also, the sufficient functions of CaMKII in LTP induction. Importantly, our LTP induction conditions (both electrically and

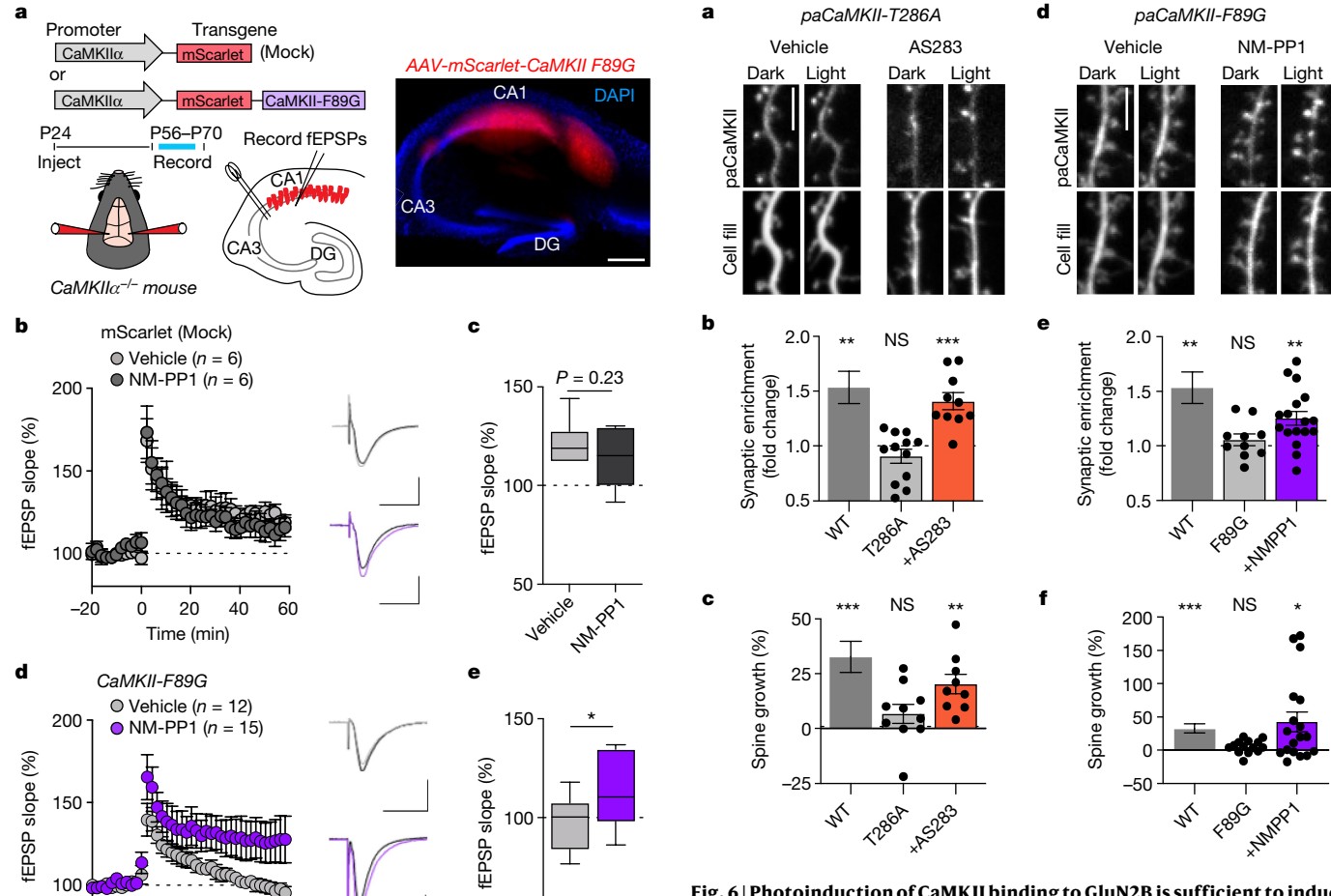

**Fig. 5 | Pharmacogenetic restoration of CaMKII binding to GluN2B restores LTP in hippocampal slices. a**, Schematic illustration of the molecular replacement approach (top and middle) and representative image of an acute hippocampal slice expressing AAV-mScarlet-CaMKII F89G (red) and stained with DAPI to label nuclei (blue; bottom). Scale bar, 500 µm. This experiment was repeated independently at least four times with similar results. **b**, fEPSPs were measured in slices from CaMKIIα KO mice injected with AAV-mScarlet (mock) after two times HFS stimulation. Slices were treated either with vehicle or 10 µM NM-PP1 for 15 min before HFS and for 5 min post-LTP induction. Scale bar, 0.5 mV by 20 ms. Data are presented as mean values ± s.e.m. **c**, Quantification of the synaptic response 60 min following LTP induction in AAV-mScarlet (mock)-expressing slices after vehicle or NM-PP1 (n = 6 slices from six animals; two-tailed Student's t-test). The box plots show the medians (centre lines) and quartiles (box limits) with Tukey whiskers. P = 0.2340. **d**, fEPSPs were measured in slices from CaMKIIα KO mice injected with AAV-mScarlet-CaMKII-F89G after two times HFS stimulation. Slices were treated either with vehicle or 10 µM NM-PP1 for 15 min before and for 5 min post-HFS. Scale bar, 0.5 mV by 20 ms. Data are presented as mean values ± s.e.m. **e**, Quantification of synaptic response 60 min following LTP induction in AAV-mScarlet-CaMKII-F89G-expressing slices after vehicle or NM-PP1 (n = 12; 15 slices from 13 animals; one-tailed Student's t-test with Welch's correction). The box plots show the medians (centre lines) and quartiles (box limits) with Tukey whiskers. P = 0.0235. *P < 0.05.

**Fig. 6 | Photoinduction of CaMKII binding to GluN2B is sufficient to induce spine growth in hippocampal neurons.** Data are presented as mean values ± s.e.m. **a**, Representative images of GFP–paCaMKII T286A and mCherry cell fill expressed in cultured hippocampal neurons before and after blue-light stimulation. Cells treated with 10 µM AS283 are marked. Scale bar, 5 µm. **b**, Fold change of paCaMKII T286A (±10 µM AS283) synaptic enrichment values 15 min after photoactivation compared with paCaMKII WT (from Fig. 1e) (n = 11, 12, 10 cells; one-sample t-test). WT P = 0.0042; T286A P = 0.1773; T286A + AS283 P = 0.0005. **P < 0.01, ***P < 0.001. **c**, Changes in dendritic spine area were measured after photoactivation of paCaMKII T286A (10 µM AS283) compared with paCaMKII WT (from Fig. 1f) (n = 12, 10, 9 cells; one-sample t-test). WT P = 0.0008; T286A P = 0.1519; T286A + AS283 P = 0.0018. **P < 0.01, ***P < 0.001. **d**, Representative images of GFP–paCaMKII F89G and mCherry cell fill expressed in cultured hippocampal neurons before and after blue-light stimulation. Cells treated with 10 µM NM-PP1 are marked. Scale bar, 5 µm. **e**, Fold change of paCaMKII F89G (10 µM NM-PP1) synaptic enrichment values 15 min after photoactivation compared with paCaMKII WT (from Fig. 1e) (n = 11, 10, 17 cells; one-sample t-test). WT P = 0.0042; F89G P = 0.2924; F89G + NM-PP1 P = 0.0010. **P < 0.01. **f**, Changes in dendritic spine area were measured after photoactivation of paCaMKII F89G (±10 µM NM-PP1) compared with paCaMKII WT (from Fig. 1f) (n = 12, 14, 18 cells; one-sample t-test). WT P = 0.0008; F89G P = 0.0537; F89G + NM-PP1 P = 0.0101. *P < 0.05, ***P < 0.001.

by light) are (1) relatively mild, (2) expected to engage the main LTP mechanisms that are studied at the hippocampal CA3–CA1 synapse and (3) similar to the protocols used previously to suggest involvement of kinase activity. Although this does not formally rule out that there could be some forms of LTP in which CaMKII also has to contribute the enzymatic kinase that is required for LTP, it does show that the essential general contribution of CaMKII to LTP does not have to involve its enzymatic activity. Notably, even though phosphorylation of the perhaps most well-known CaMKII substrate site on another protein, S831 on the AMPA-type glutamate receptor subunit GluA1, should promote LTP (as it increases single-channel conductance)[32,33], there has been a strong previous indication that this phosphorylation is not required for LTP[34–36]. Other CaMKII phosphorylation sites that may contribute to LTP include S277 and S281 on TARPγ-8, an auxiliary

protein that aids in AMPA-type glutamate receptor trafficking and function[37]. However, even though mutation of these sites reduced LTP, the increase in their phosphorylation after cLTP stimuli was modest, and the level of basal phosphorylation was high[37]. Indeed, another study found that mutating these sites on TARPγ-8 reduced basal transmission but not LTP[38]. Furthermore, although the modest increase in TARPγ-8 phosphorylation after cLTP stimuli was blocked by KN93 (ref. 37), this CaMKII inhibitor also inhibits synaptic PKC[39], another kinase known to phosphorylate TARPγ-8. Thus, even though TARPγ-8 phosphorylation may potentiate synaptic transmission, the basal phosphorylation may be sufficient, and the modest LTP-induced increase may not even require CaMKII. In contrast to phosphorylation of the exogenous substrate sites on GluA1 and TARPγ-8, the autophosphorylation of CaMKII at T286 is clearly required for LTP[8]. However, according to our results, the function of pT286 in LTP is not for the generation of the $Ca^{2+}$-independent 'autonomous' activity[15,16] as generally proposed[1-3] but instead, to enable the required regulation of synaptic CaMKII accumulation and the coinciding structural CaMKII functions. Thus, generation of pT286 is part of the initial upstream signal processing that leads to LTP induction, which enables the downstream output through the structural functions of CaMKII without any requirement for enzymatic activity toward exogenous substrates. Notably, the fact that LTP can be induced and then maintained in the T286A mutant mice with the help of the AS283 inhibitor (which inhibits enzymatic activity while promoting GluN2B binding) also indicates that LTP maintenance does not require the autonomous activity generated by pT286 either. Indeed, the complex biochemical regulation of the pT286 reaction (that includes dual requirement for CaM and crossregulation with other autophosphorylation reactions) much favours its participation in signal processing rather than LTP maintenance[40-42]. Whereas structural CaMKII functions are clearly central to LTP induction, the underlying mechanisms remain to be elucidated. Ultimately, any structural LTP mechanism has to regulate the F-actin cytoskeleton[43,44]. However, even though direct F-actin binding and bundling are among the structural functions of CaMKII, this is largely restricted to the CaMKIIß isoform rather than the major α isoform studied here[44-47]. Moreover, binding to F-actin versus GluN2B is oppositely regulated, with activating signals disrupting the F-actin binding rather than inducing it[44-47]. However, intriguing alternative possibilities are offered by recent findings showing that regulated CaMKII interactions with GluN2B can lead to liquid–liquid phase separation that can rearrange other post-synaptic proteins[48-50], possibly providing an emerging additional organizing principle at excitatory synapses. Additionally, it has not escaped our notice that the specific findings we have reported immediately suggest possible therapeutic applications for chronic CaMKII inhibition.

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

## Methods

### Experimental animals

All animal procedures were approved by the University of Colorado Institutional Animal Care and Use Committee and carried out in accordance with the National Institutes of Health best practices for animal use. The University of Colorado Anschutz Medical Campus is accredited by the Association for Assessment and Accreditation of Laboratory Animal Care, International. All animals were housed in ventilated cages on a 12-h light/12-h dark cycle and were provided ad libitum access to food and water. Male WT and T286A knock-in mice (on a C57BL/6 background) from heterozygous breeder pairs (8–12 weeks old) were used for slice electrophysiology. Mixed sex CaMKII knockout (KO) mice were used for AAV injections and slice electrophysiology. Mixed sex pups from Sprague–Dawley rats (Charles River) or individual pups from heterozygous breeding of the GluN2B$^{\Delta CaMKII}$ mutant mice were used to prepare dissociated hippocampal cultures for imaging. The mutant mice used here were described previously. The CaMKIIα knockout line used here was made in house[51], the GluN2B$^{\Delta CaMKII}$ line was provided by Johannes Hell[22,24] and the T286A line was provided by Ryohei Yasuda with permission from Karl Peter Giese[8].

### Material and DNA constructs

Material was obtained from Sigma, unless noted otherwise. CMV-mEGFP(A206K)-paCaMKII was a gift from Hideji Murakoshi (Addgene plasmid 165438). The pAAV-CaMKIIα-mScarlet vector was a gift from Karl Deisseroth (Addgene 131000). CaMKIIα-F89G complementary DNA was cloned into the multiple cloning site after mScarlet using BsrGI and EcoRV.

### Protein purification

Expression and purification of CaMKIIα, CaM, GST–GluN2Bc and GST–GluA1 were conducted according to the established protocol described in detail previously[9,42,52,53]. CaMKIIα was purified from a baculovirus/Sf9 cell expression system. CaM and GST–GluN2Bc WT and mutant constructs were purified from BL21 bacteria.

### Immunoblot analysis

Protein concentration was determined using the Pierce BCA protein assay (Thermo-Fisher). Before undergoing SDS–PAGE, samples were boiled in Laemmli sample buffer for 5 min at 95 °C. Proteins were separated in a resolving phase polymerized from 10% acrylamide and then transferred to a polyvinylidene difluoride membrane at 24 V for 1–2 h at 4 °C. Membranes were blocked in 5% milk or bovine serum albumin (BSA) and incubated with anti-CaMKIIα (1:4,000, CBα2; available at Invitrogen but made in house), anti-CaMKIIα (1:2,000; BD), pT286-CaMKII (1:2,500; Phospho-Solutions), anti-GST (1:2,000; Millipore), pS831-GluA1 (1:2,000; Phospho-Solutions) and pS1303 (1:2,000; Millipore) followed by either Amersham ECL goat anti-mouse or anti-rabbit secondary linked to horseradish peroxidase (1:10,000; GE Healthcare). Blots were developed using chemiluminescence (Super Signal West Femto; Thermo-Fisher) imaged using the Chemi-Imager 4400 system (Alpha-Innotech) or imaged directly by fluorescence (Cytiva CyDye 700 goat anti-mouse (1:10,000) and CyDye 800 goat anti-rabbit (1:10,000) secondary antibodies) using an OdysseyFc imaging instrument. All immunoblots were analysed by densitometry (ImageJ; v.2.9.0/1.53t). Phosphosignal was corrected to total protein. Relative band intensity was normalized as a percentage of control conditions on the same blot whenever possible. Replicates of controls were included on each blot to allow for comparison between multiple experiments when multiple blots were required (although technical replicates of controls were not included as multiple '$n$'). Uncropped and unprocessed western blot images are provided in the Supplementary Information.

### In vitro phosphorylation assays

CaMKII-mediated phosphorylation of GluA1 S831 or GluN2B S1303 was measured by in vitro kinase reaction with a purified GST–GluA1 C-terminal tail or GST-GluN2B C-terminal tail. Reactions contained 40 nM CaMKII, 1 μM GST–GluA1, 50 mM PIPES, pH 7.1, 2 mM CaCl$_2$, 10 mM MgCl$_2$, 1 μM calmodulin, 1–4 mM ATP and 1 μM okadaic acid. Negative controls were prepared without ATP. Reactions were done at 30 °C for 20 s; reactions were stopped by adding SDS loading buffer and incubation in a boiling water bath for 5 min. GST, phospho-S831, CaMKII, phospho-T286 or phospho-S1303 were detected in the samples by immunoblot analysis.

### $K_m$ and $K_i$ determination

The inhibition constant $K_i$ was calculated using the Cheng–Prusoff equation for competitive inhibition: $K_i = IC50/(1 + [ATP]/K_m)$. Using Michaelis–Menten analyses, the CaMKII $K_m$ for ATP at the described experimental conditions was experimentally determined to be $K_m ATP = 33.3$ μM (Extended Data Fig. 2b). This is within the range of previously published $K_m$ values (8–127 μM)[26,54–56].

### CaMKII binding to GluN2B in vitro

CaMKII–GluN2B binding assays were done as described[9,13,25]. Briefly, GST–GluN2B-c tail (GST–2BC) was immobilized on anti-GST antibody-coated microtiter plates (Thermo Scientific), blocked for 30 min with 5% BSA and then overlaid with 40 nm CaMKII (subunit concentration) in PIPES-buffered saline (pH 7.2) containing 2 mM Ca$^{2+}$, 1 μM CaM, 1 mM Mg$^{2+}$, 0.1% BSA and 0.1% Tween-20 for 20 min at room temperature. The addition of 1 mM ADP, 1 mM ATP, 10 μM AS283 (or AS105) or 10 μM NM-PP1 was added accordingly (Figs. 1–6). After extensive washes in buffer containing 1 mM EGTA, GST–2BC and bound CaMKII were eluted for 10 min in SDS loading buffer at 95 °C. Bound CaMKII was measured via immunoblot.

### Cell culture of HEK293 cells

Human embryonic kidney cells (HEK293; authenticated by short tandem repeat analysis) were cultured in Dulbecco's modified Eagle's medium (Gibco) supplemented with 10% foetal bovine serum (Sigma) and 1% penicillin–streptomycin solution (Gibco). HEK cells were not tested for mycoplasma. HEK cells were grown on 10 cm culture flasks and split every 3–4 days (at approximately 90% confluency). Over-expression of CaMKII WT or mutants for use with in vitro assays was performed as previously described[14,42]. For imaging experiments, cells were split into 12-well culture dishes on 18 mm no. 1 glass coverslips.

### Live imaging of HEK cells

HEK cells were grown and transfected with expression vectors for GFP–CaMKII mutants and pDisplay-mCh-GluN2B-c tail (2BC) as previously described[24,57]; the GluN2B construct used included an S1303A mutation to avoid potential complications by differential phosphorylation of this regulatory site[13,27] in the different conditions with or without kinase inhibitors. GFP–CaMKII colocalization with GluN2B in response to a Ca$^{2+}$ stimulus induced by 10 μM ionomycin was monitored for 5–10 min at 32 °C in imaging buffer (0.87× Hanks Balanced Salt Solution, 25 mM HEPES, pH 7.4, 2 mM glucose, 2 mM CaCl$_2$, 1 mM MgCl$_2$) by fluorescence microscopy. Images were acquired on a Zeiss Axiovert 200M equipped with a climate control chamber using SlideBook software (Intelligent Imaging Innovations; v.6.0). Colocalization analysis was performed by calculating the Pearson's correlation (correlation index) with ImageJ of pDisplay-mCh-2BC and GFP–CaMKII within the cytoplasm of HEK cells after background subtraction.

### paCaMKII stimulation in HEK cells

HEK cells were transfected with GFP–paCaMKII (WT, K42M and I205K) and pDisplay-mCherry-2BC$^{S1303A}$ and left in the dark for 10 min before

photoactivation to ensure that paCaMKII was in the dark state. GFP–paCaMKII was then photoactivated and imaged simultaneously via confocal imaging over a 3 µm Z stack (step size: 0.6 µm) or a single plane with 488 nm excitation once per minute for a total of 5 min. The correlation index was measured the same as for ionomycin-induced colocalization.

## Primary hippocampal culture preparation

To prepare primary rat hippocampal neurons, hippocampi were dissected from mixed sex rat pups (P0), dissociated in papain for 1 h and plated at 100,000 cells per well on 18 mm no. 1 glass coverslips in 12-well culture dishes for imaging and 500,000 cells per well on 6-well culture dishes for biochemistry in plating media: Minimal Essential Media (Gibco) containing 10% foetal bovine serum (Sigma) and 1% penicillin–streptomycin (Gibco). Mouse hippocampi were dissected from individual mouse pups (P1–P2), dissociated in papain for 30 min and plated at 200,000–250,000 cells ml$^{-1}$ on glass coverslips for imaging. For all cultures, plating media were replaced on day in vitro (DIV) 1 with feeding media: Neurobasal A (Gibco) containing 2% B27 (Gibco) and 1% Glutamax (Sigma). On DIV 7, half of conditioned feeding media was replaced with fresh feeding media containing 2% 5-Fluoro-2′-deoxyuridine (Sigma). At DIV 14–18, neurons were transfected with 1 µg total complementary DNA per well using Lipofectamine 2000 (Invitrogen) and then imaged 2–3 days later.

## cLTP stimulation

cLTP was induced with 100 µM glutamate and 10 µM glycine for 45 s. Treatments were followed by washout in 5 volumes of fresh artificial cerebral spinal fluid (ACSF). For biochemistry experiments, neurons were treated with 1 µM tetrodotoxin to silence neurons before treatment with cLTP.

## paCaMKII stimulation in hippocampal neurons

Neurons were wrapped in aluminium foil immediately following transfection and only exposed to red light in imaging. One image was then taken of each neuron to serve as a pre-photoactivation baseline. Immediately following this baseline image, paCaMKII was globally photoactivated with a 405 nm laser pulse (100 ms exposure, 75% laser power) once every 10 s for a total of 60 s. Neurons were then imaged 15 min after stimulation and assessed for CaMKII synaptic enrichment and dendritic spine growth.

## Image analysis of hippocampal neurons

Neuronal cultures were transfected on DIV 15–18 and imaged 24–48 h later. All experiments with overexpression of CaMKII mutants for cLTP stimulation were co-transfected with a short hairpin RNA directed against the CaMKII 5′ untranslated region to knock down endogenous CaMKII. Images were collected at 32 °C in HEPES buffered imaging solution containing 130 mM NaCl, 5 mM KCl, 10 mM HEPES, pH 7.4, 20 mM glucose, 2 mM CaCl$_2$ and 1 mM MgCl$_2$. Images of individual neurons from two independent cultures were acquired by 0.5 µm steps over 6 µm. Two-dimensional maximum-intensity projection images were then generated and analysed using a custom-build programme in ImageJ. The programme utilizes combinatorial thresholding to mask regions of the cell that contain high-intensity puncta of post-synaptic density protein 95 (PSD95; a marker of the post-synaptic side of excitatory synapses in dendritic spines) and regions of the dendritic shaft that contain no fluorescence intensity of PSD95. As a measure of synaptic enrichment, the ratio of mean CaMKII fluorescence intensity of the PSD95 mask to the mean CaMKII fluorescence intensity in the dendritic shaft mask is measured. Spine growth was assessed by measuring the changes in mCherry cell fill fluorescence intensity within dendritic spine divided by the initial fluorescence intensity (F/F0).

## AAV production

AAV vectors were constructed from an empty AAV transfer plasmid where the expression cassette is as follows: left inverted terminal repeats (ITR), CaMKIIα promoter, mScarlet, multiple cloning site, woodchuck hepatitis virus (WHP) posttranslational element (WPRE) and right ITR. To generate AAVs, HEK293T cells were transfected with an AAV transfer plasmid, pHelper and pRC-DJ. AAVs were purified as previously described[58]. Briefly, 72 h post-transfection, cells were harvested and lysed, and virus was purified on an iodixanol gradient via ultracentrifugation. Virus was harvested from the 40% fraction; then, it was concentrated and washed in a 100 K molecular weight cut-off (MWCO) Amicon filter. AAVs were titered by infecting mouse hippocampal cultures with serial dilutions and used for stereotactic infections at $1 \times 10^9$ infections units µl$^{-1}$. The following AAVs were used: AAV$_{DJ}$-CaMKIIα-mScarlet-CaMKII$^{F89G}$ and AAV$_{DJ}$-CaMKIIα-mScarlet.

## Stereotactic surgeries

Stereotactic injections were performed on P24 CaMKIIα KO mice. Animals were anesthetized with an intraperitoneal injection of 2,2,2-Tribromoethanol (250 mg kg$^{-1}$) and then head fixed to a stereotactic frame (David Kopf Instruments). AAVs (0.2–0.5 µl) were injected into intermediate CA1 at a rate of 10 ml h$^{-1}$ using a syringe pump (World Precision Instruments). Coordinates (in millimetres) were anterior–posterior: −3.17; mediolateral: ±3.45 (relative to Bregma); and dorsoventral: −2.5 (relative to pia). To confirm specificity of the injection site following LTP recordings, slices were fixed in 4% paraformaldehyde; then, they were coverslipped and imaged on an Olympus slide scanning microscope.

## Hippocampal slice preparation

WT and mutant mouse hippocampal slices were prepared using P56–P70 mice. Isoflurane-anesthetized mice were rapidly decapitated, and the brain was dissected in an ice-cold high-sucrose solution containing 220 mM sucrose, 12 mM MgSO$_4$, 10 mM glucose, 0.2 mM CaCl$_2$, 0.5 mM KCl, 0.65 mM NaH$_2$PO$_4$, 13 mM NaHCO$_3$ and 1.8 mM ascorbate. Transverse hippocampal slices (400 µm) were made using a tissue chopper (McIlwain) and transferred into 32 °C ACSF containing 124 mM NaCl, 3.5 mM KCl, 1.3 mM NaH$_2$PO$_4$, 26 mM NaHCO$_3$, 10 mM glucose, 2 mM CaCl$_2$, 1 mM MgSO$_4$ and 1.8 mM ascorbate. All slices were recovered in 95% O$_2$/5% CO$_2$ for at least 1.5 h before experimentation.

## Extracellular field recordings

For electrophysiological slice recording experiments, a glass micropipette (typical resistance of 0.4–0.8 megaohm when filled with ACSF) was used to record fEPSPs from the CA1 dendritic layer in response to stimulation of the Schaffer collaterals using a tungsten bipolar electrode. Slices were continually perfused with 30.5 °C ± 0.5 °C ACSF at a rate of 2.5 ± 0.5 ml min$^{-1}$ during recordings. Stimuli were delivered every 20 s, and three responses (1 min) were averaged for analysis. Data were analysed using WinLTP software with slope calculated as the initial rise from 10 to 60% of the response peak. Input–output curves were generated by increasing the stimulus intensity at a constant interval until a maximum response or population spike was noted to determine stimulation that elicits 50% of maximum slope. Paired-pulse recordings (50 ms interpulse interval) were acquired from 50% max slope, and no differences in pre-synaptic facilitation were seen in mutant slices. A stable baseline was acquired for a minimum of 20 min at 50% maximum slope before high-frequency stimulation (HFS; 2 × 100 Hz, 10 s interval) was applied. Responses were recorded for 40 or 60 min after HFS. Change in slope was calculated as a ratio of the average slope of the 20 min baseline (before HFS). Bar graphs of the percentage of fEPSP slope were calculated by averaging the last 10 min time points post-HFS and normalized to baseline.

## Statistical analysis

All data are shown as mean ± s.e.m. Statistical significance is indicated in the figure legends. Statistics were performed using Prism (Graph-Pad) software. Imaging experiments were obtained using SlideBook 6.0 software and analysed using ImageJ. Immunoblots were analysed by densitometry using ImageJ. All data were tested for their ability to meet parametric conditions, as evaluated by a Shapiro–Wilk test for normal distribution and a Brown–Forsythe test (three or more groups) or an $F$ test (two groups) to determine equal variance. All comparisons between two groups met parametric criteria, and independent samples were analysed using unpaired Student's $t$-tests. Comparisons between three or more groups meeting parametric criteria were done by one- or two-way ANOVA with specific post hoc analysis as indicated in the figure legends. Sample sizes for this study were determined based on previous experience; post hoc power analysis was conducted, confirming that our studies were adequately powered to detect statistical significance of effects. Biological replication was achieved by measuring each unique cell, sample and hippocampal slice once, derived from at least two separate cultures and a minimum of four distinct wells. Randomization was accomplished by reversing the sample order for every experiment. Investigators were not blinded to the samples during collection or analysis. Blinding was not performed owing to resources constraints combined with the nature of experiments and analysis having low potential for introducing bias.

## Reporting summary

Further information on research design is available in the Nature Portfolio Reporting Summary linked to this article.

## Data availability

The data used in our analysis can be found at Mendeley Data[59]. Source data are provided with this paper.

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

**Acknowledgements** The calmodulin-dependent protein kinase II inhibitor AS100283 (here referred to as AS283) was developed by Allosteros Therapeutics, Inc., and a stock supply was synthesized for distribution to the scientific community by the generosity of the Leducq Foundation, Boston. The construct for photoactivatable paCaMKII was a gift from Hideji Murakoshi. This work was supported by the National Institutes of Health (grants T32 GM007635 (supporting J.E.T. and C.N.B.), F31 AG069458 (to N.L.R.), F31 MH125510 (to E.E.B.), F31 NS129254 (to C.N.B.), R01 MH116901 (to J.A.), R01 NS040701 (to M.L.D), R01 NS110383 (to M.L.D. and K.U.B), R01 NS081248 (to K.U.B.), R01 NS118786 (to K.U.B.) and R01 AG067713 (to K.U.B.)).

**Author contributions** J.E.T, M.E.L, N.L.R., R.K.F., E.E.B., C.N.B. and S.J.C performed experiments and analyses. Specifically, J.E.T. performed the characterization of the calmodulin-dependent protein kinase II (CaMKII) inhibitors and F89G mutation on kinase activity, GluN2B binding and movement to synapses. M.E.L. performed all experiments with paCaMKII in HEK cells or neurons. N.L.R. and R.K.F. performed the electrophysiological recordings. E.E.B. performed the stereotactic injections. C.N.B. performed the biochemical experiment with paCaMKII. S.J.C. provided help and guidance with the biochemical assessment of CaMKII activity and GluN2B binding. H.S., J.A. and M.L.D. provided essential material or experimental setups and advice. J.E.T. and K.U.B. conceived this study, with contributions from M.E.L. and N.L.R., and wrote the initial draught; all authors contributed to the final manuscript.

**Competing interests** The Regents of the University of Colorado have filed a provisional patent application, with inventors J.E.T., M.E.L., N.L.R. and K.U.B., capturing the findings reported in this manuscript. K.U.B. is a co-founder and board member of Neurexis Therapeutics, a company that seeks to develop a CaMKII inhibitor into a therapeutic drug for cerebral ischaemia. The other authors declare no competing interests.

**Additional information**
**Correspondence and requests for materials** should be addressed to K. Ulrich Bayer.

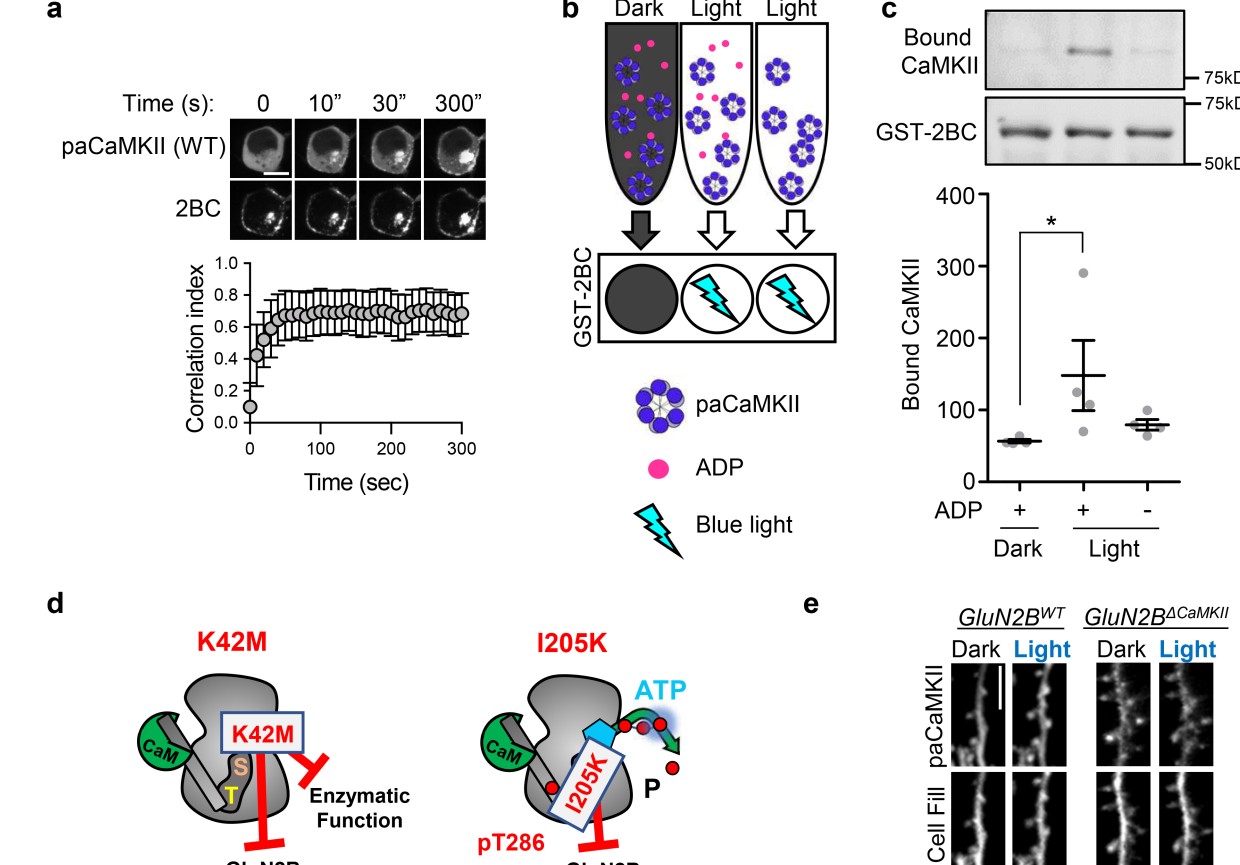

**Extended Data Fig. 1 | paCaMKII photoactivation-induced sLTP requires GluN2B binding. Data are presented as mean values +/− SEM. a,** Representative images of GFP-paCaMKII WT and mCherry-2BC expressed in cultured hippocampal neurons before and after blue light stimulation. Time course of correlation indices of GFP-paCaMKII and mCh-2BC after photoactivation. Images were acquired in a single plane ever 10 s (n = 4 cells). Scale bar indicates 10 μm. **b,** Schematic of blue light-induced paCaMKII pulldown to immobilized GST-2BC. **c,** Binding of CaMKII-F89G to immobilized GST-2BC was induced by blue light in the presence of either 100 ADP μM alone or without nucleotide. (n = 4 independent samples; *p < 0.05; one-way Kruskal Wallis with Dunn's comparison test). Dark:Light+ADP p = 0.0132; Dark:Light-ADP p = 0.1860; Light+ADP:Light-ADP p = 0.9779. **d,** Schematic of how K42M and I205K CaMKII mutations affect enzymatic activity and the induction of GluN2B binding. **e,** Representative images of dissociated mouse WT and GluN2B^ΔCaMKII hippocampal cultures expressing GFP-paCaMKII WT and mCherry cell fill before and after photoactivation. Scale bar indicates 5 μm. n = 9, 9 cells; See Fig. 1f, g (right) for quantification.

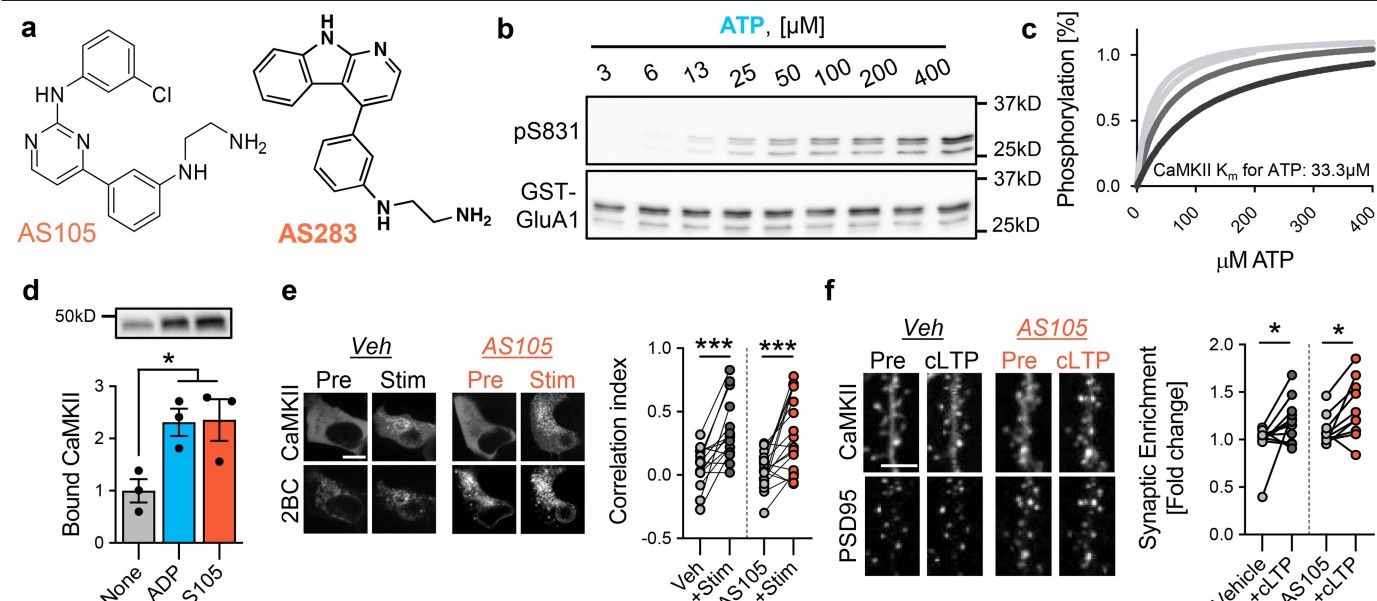

**Extended Data Fig. 2 | AS105 inhibits CaMKII enzymatic function but does not impair GluN2B binding or synaptic localization. Data are presented as mean values** +/− **SEM. a**, Comparison of AS105 vs AS283 structure. **b**, Representative Immunoblots and quantification of *in vitro* kinase reactions at 30 °C with increasing ATP concentration measuring purified CaMKIIα phosphorylation of GST-GluA1 S831. Quantification shown in panel c. **c**, The Km value of CaMKII for ATP of 33.3 µM was derived from the concentration of ATP that achieved a half-maximal response in 4 independent experiments (95% confidence interval: 4.59 to 52.06). **d**, Binding of purified CaMKIIα to immobilized GST-GluN2B-c was induced by Ca²⁺/CaM without nucleotide, or in the presence of either 4 mM ADP or with 10 µM AS105 (n = 3 independent samples; *p < 0.05; one-way ANOVA with Tukey's multiple comparisons test). −:ADP p = 0.0394;

−:AS105 p = 0.0349. **e**, Confocal microscopy images of HEK293 cells co-expressing GFP-CaMKII and mCherry-2BC. Correlation indices before and after ionomycin-induced colocalization of GFP-CaMKII and mCherry-2BC under vehicle or 10 µM AS105 conditions (n = 23,18 cells; ***p < 0.001, RM two-way ANOVA with Šídák's multiple comparisons test). Scale bar indicates 10 µm. Vehicle p < 0.0001; AS105 p < 0.0001. **f**, Confocal microscopy images of dissociated rat hippocampal cultures expressing GFP-CaMKII and mCh-PSD95 intrabody in the presence of 10 µM AS105 before and after cLTP. CaMKII synaptic enrichment was measured before and after cLTP (n = 11,10 cells; *p < 0.05; RM two-way ANOVA with Šídák's multiple comparisons test). Scale bar indicates 5 µm. Vehicle p = 0.0376; AS105 p = 0.0330.

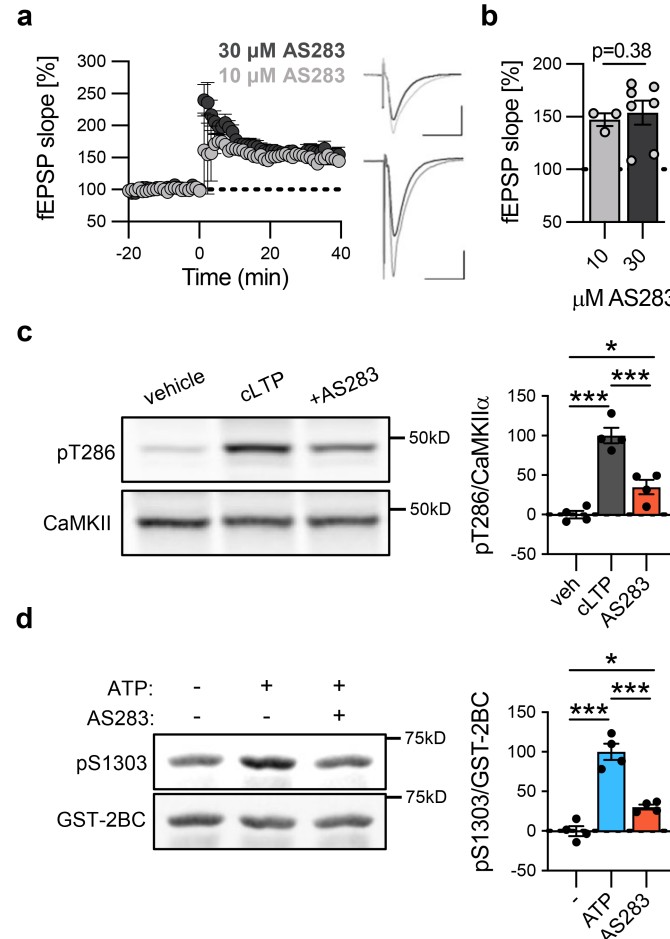

**Extended Data Fig. 3 | No difference in LTP in WT mice when CaMKII is inhibited by either 10 or 30 μM AS283. Data are presented as mean values +/− SEM. a**, 2x HFS potentiates the CA3-CA1 Schaffer collateral pathway in WT mice when CaMKII enzymatic activity is inhibited by 10 or 30 μM AS283 (incubated for 15 min prior to HFS, and washed out 5 min after). **b**, Quantification of LTP in WT mice after 10 or 30 μM AS283 (n = 3,7 hippocampal slices; Mann-Whitney test). p = 0.3833. **c**, Representative immunoblot and quantification of CaMKII T286 phosphorylation in dissociated hippocampal cultures after cLTP and inhibition by 10 μM AS283 (n = 4 independent cell preparations; one-way ANOVA, Tukey's multiple comparisons test). The inhibition of T286p in seen hippocampal cultures matches the inhibition seen *in vitro* (see Fig. 2b). Vehicle:cLTP p < 0.0001; Vehicle:cLTP+AS283 p = 0.0368; cLTP:cLTP+AS283 p = 0.0009. **d**, Representative immunoblot and quantification of *in vitro* kinase reactions at 30 °C with purified CaMKII and GST-2BC measuring S1303 phosphorylation and inhibition by 10 μM AS283 (n = 4 independent samples; one-way ANOVA, Tukey's multiple comparisons test). −:ATP p < 0.0001; −:ATP+AS283 p = 0.0363; ATP:ATP+AS283 p = 0.0002. The inhibition seen for S1303 here matches the inhibition seen for T286 in hippocampal neurons in panel c.

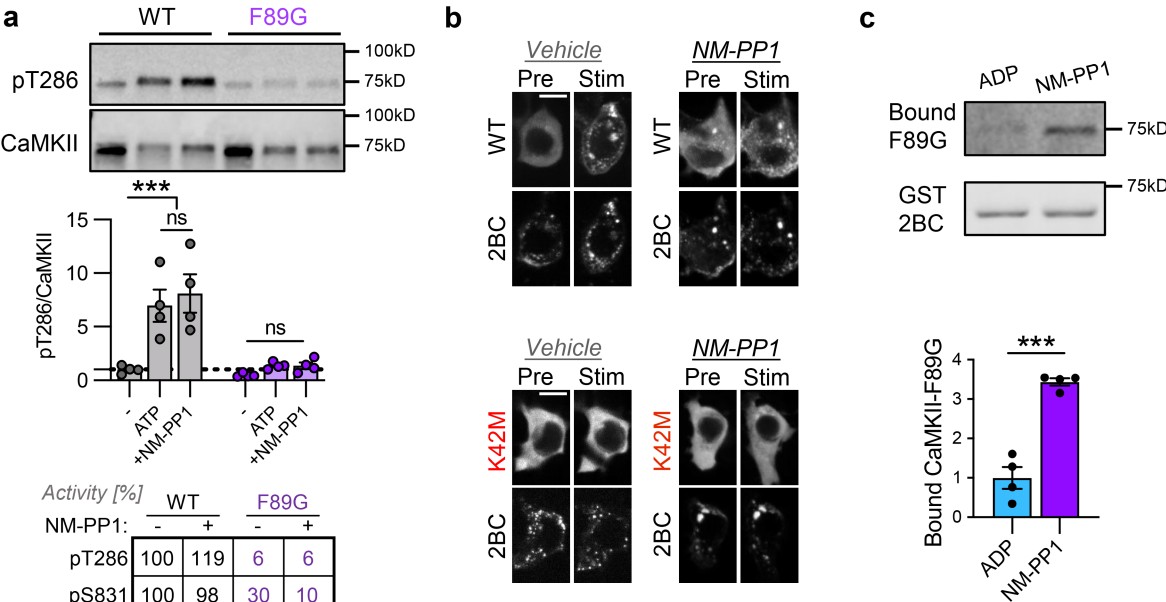

Activity [%]

| NM-PP1: | WT | | F89G | |
|---|---|---|---|---|
| | − | + | − | + |
| pT286 | 100 | 119 | 6 | 6 |
| pS831 | 100 | 98 | 30 | 10 |

**Extended Data Fig. 4 | CaMKII WT and K42M are unaffected by NM-PP1 in HEK cells.** Data are presented as mean values +/− SEM. **a**, Representative immunoblot and quantification of *in vitro* kinase reactions at 30 °C in 1 mM ATP measuring CaMKIIα WT and F89G phosphorylation of T286 (n = 4 independent samples, **p < 0.01, ***p < 0.001; two-way ANOVA with Šídák's multiple comparisons test). WT) −:ATP p = 0.0012; −:ATP+NM-PP1 p = 0.0002; ATP:ATP+NM-PP1 p = 0.7989. F89G) −:ATP p = 0.8920; −:ATP+NM-PP1 p = 0.8946; ATP:ATP+NM-PP1 p > 0.9999. **b**, Confocal microscopy images of HEK293 cells co-expressing GFP-CaMKII WT (top) or GFP-CaMKII K42M (bottom) and mCherry-2BC in the presence of vehicle or 10 μM NM-PP1 before and after ionomycin stimulation. Scale bar indicates 10 μm. n = 22, 17 cells; see Fig. 4c for quantification. **c**, Binding of CaMKII-F89G to immobilized GST-GluN2B-c was induced by Ca²⁺/CaM in the presence of either 1 mM ADP or 10 μM NM-PP1. The wash buffer also contained AD or NM-PP1, Ca²⁺/CaM was replaced with EGTA (n = 4 independent samples; ***p < 0.001; two-tailed student's t-test). p = 0.0002.

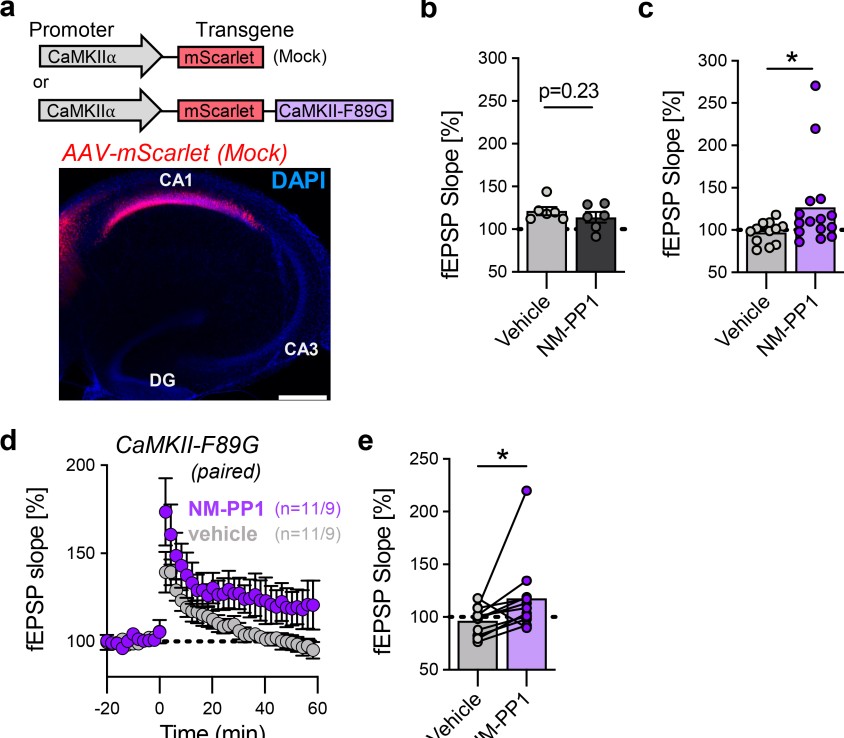

**Extended Data Fig. 5 | Pharmacogentic restoration of CaMKII binding to GluN2B enables LTP in hippocampal slices.** Data are presented as mean values +/− SEM. **a**, Schematic illustration of the molecular replacement approach (top) and representative image of acute hippocampal slice expressing AAV-mScarlet (Mock) (red) and labelled with DAPI (blue) (bottom). Scale bar indicates 500 μm. This experiment was repeated independently at least 4 times with similar results. **b**, Representation of the data shown in Fig. 5c as a scatter plot. (n = 6 slices; p = 0.2340, two-tailed student's t-test). **c**, Representation of the data shown in Fig. 5e as a scatter plot (n = 12,15 slices; *p < 0.05, one-tailed student's

t-test with Welch's correction). p = 0.0235. Significant even with removal of two higher data points for F89G + NM-PP1 (p = 0.0320; one-tailed student's t-test). **d**, fEPSPs were measured in slices from CaMKIIα KO mice injected with AAV-mScarlet-CaMKII-F89G after 2x HFS stimulation. The paired vehicle and NM-PP1-treated slices that were available from the same mouse were paired in statistical comparison. **e**, Quantification of synaptic response 60 min following LTP induction in AAV-mScarlet-CaMKII-F89G-expressing paired slices from the same animal that were treated either with vehicle or NM-PP1 (n = 11 slices from 9 mice; *p < 0.05, one-tailed student's t-test). p = 0.0304.

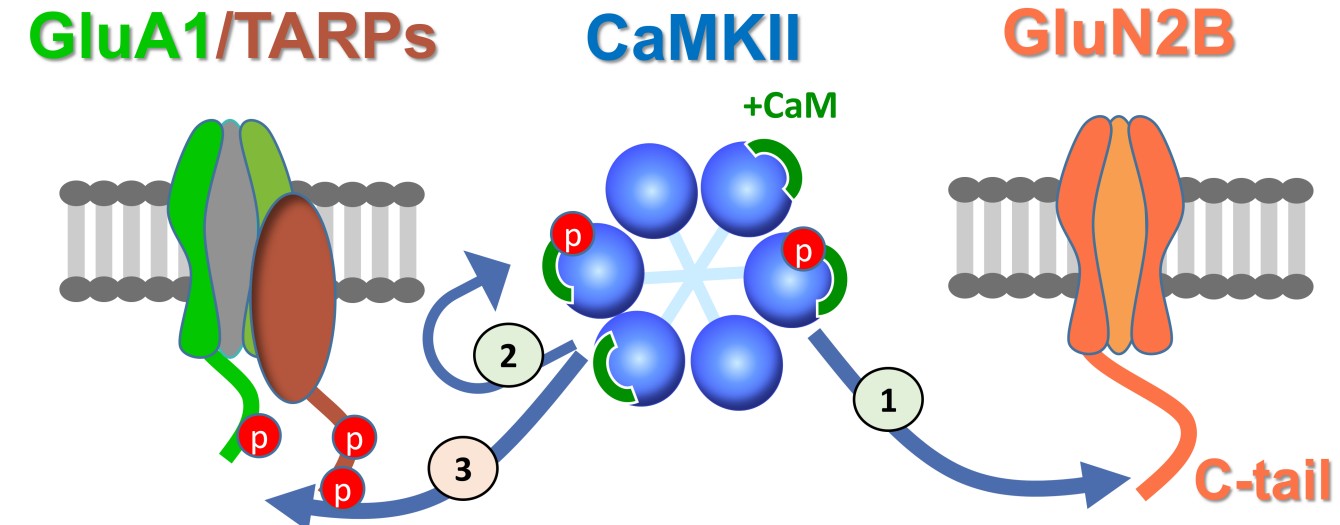

**Extended Data Fig. 6 | Schematic of primary conclusions.** Illustration of CaMKII enzymatic and structural functions and their roles in LTP induction. (1) CaMKII structural functions are required and sufficient to induce LTP. (2) CaMKII autophosphorylation at T286 is required to induce structural functions, but can be circumvented with ATP-competitive inhibitors that enhance GluN2B binding. (3) CaMKII phosphorylation of external substrates is not required to induce LTP.

**Extended Data Table 1 | Selectivity of AS283 compared to AS105**

| | AS100105 / Scios | AS100283 / Tricyclic |
|---|---|---|
| Arg(h) | 87 | 96 |
| AMPKα1(h) | 38 | 70 |
| Aurora-C(h) | 57 | 63 |
| CaMKI(h) | 83 | 107 |
| CaMKIIβ(h) | 6 | 5 |
| CaMKIIγ(h) | 2 | 2 |
| CaMKIIδ(h) | 2 | 2 |
| CaMKIIα(h) | | 3 |
| CaMKIV(h) | 101 | 90 |
| CDK2/cyclinE(h) | 2 | 41 |
| CDK5/p35(h) | 1 | 33 |
| cKit(h) | 91 | 50 |
| FAK(h) | 75 | 68 |
| FGFR1(h) | 91 | 58 |
| *Flt3(h)* | *2* | *3* |
| GSK3α(h) | 6 | 82 |
| GSK3β(h) | 11 | 87 |
| IGF-1R(h) | 121 | 110 |
| JAK2(h) | 134 | 121 |
| JAK3(h) | 89 | 92 |
| JNK2α2(h) | 79 | 82 |
| KDR(h) | 28 | 15 |
| MAPK1(h) | 105 | 102 |
| MAPK2(h) | 106 | 97 |
| MEK1(h) | 111 | 85 |
| mTOR(h) | 102 | 115 |
| p70S6K(h) | 77 | 18 |
| PAK2(h) | 104 | 76 |
| PDGFRβ(h) | 80 | 111 |
| PDK1(h) | 109 | 78 |
| Pim-1(h) | 69 | 50 |
| Pim-3(h) | 109 | 97 |
| PKBγ(h) | 76 | 42 |
| PKCα(h) | 51 | 49 |
| PKCδ(h) | 73 | 39 |
| PKG1α(h) | 61 | 48 |
| PKG1β(h) | 68 | 38 |
| Ret(h) | 55 | 44 |
| ROCK-II(h) | 43 | 77 |
| SAPK2a(h) | 99 | 95 |
| SGK(h) | 97 | 86 |
| PI3 Kinase (p110a/p85a)(h) | 96 | 100 |

Both inhibitors were used at 1µM. Indicated is the residual kinase activity in % of maximal activity. Compared to AS105, the selectivity of AS283 is further increased. Notably, the one remaining strongly cross-inhibited kinase (Flt3) is not significantly expressed in hippocampus. Two other kinases (KDR and p70S6K) remain somewhat inhibited by AS283, but for a block comparable to CaMKII activity, higher concentrations would be required.

# Reporting Summary

## Statistics

For all statistical analyses, confirm that the following items are present in the figure legend, table legend, main text, or Methods section.

| n/a | Confirmed | |
|---|---|---|
| ☐ | ☒ | The exact sample size (*n*) for each experimental group/condition, given as a discrete number and unit of measurement |
| ☐ | ☒ | A statement on whether measurements were taken from distinct samples or whether the same sample was measured repeatedly |
| ☐ | ☒ | The statistical test(s) used AND whether they are one- or two-sided *Only common tests should be described solely by name; describe more complex techniques in the Methods section.* |
| ☐ | ☒ | A description of all covariates tested |
| ☐ | ☒ | A description of any assumptions or corrections, such as tests of normality and adjustment for multiple comparisons |
| ☐ | ☒ | A full description of the statistical parameters including central tendency (e.g. means) or other basic estimates (e.g. regression coefficient) AND variation (e.g. standard deviation) or associated estimates of uncertainty (e.g. confidence intervals) |
| ☐ | ☒ | For null hypothesis testing, the test statistic (e.g. *F*, *t*, *r*) with confidence intervals, effect sizes, degrees of freedom and *P* value noted *Give P values as exact values whenever suitable.* |
| ☒ | ☐ | For Bayesian analysis, information on the choice of priors and Markov chain Monte Carlo settings |
| ☒ | ☐ | For hierarchical and complex designs, identification of the appropriate level for tests and full reporting of outcomes |
| ☐ | ☒ | Estimates of effect sizes (e.g. Cohen's *d*, Pearson's *r*), indicating how they were calculated |

*Our web collection on statistics for biologists contains articles on many of the points above.*

## Software and code

Policy information about availability of computer code

| Data collection | Slidebook (Intelligent Imaging Innovations [3i], Version 6.0) |
|---|---|
| Data analysis | ImageJ (Version: 2.9.0/1.53t) |

For manuscripts utilizing custom algorithms or software that are central to the research but not yet described in published literature, software must be made available to editors and reviewers. We strongly encourage code deposition in a community repository (e.g. GitHub). See the Nature Portfolio guidelines for submitting code & software for further information.

## Data

Policy information about availability of data

All manuscripts must include a data availability statement. This statement should provide the following information, where applicable:

- Accession codes, unique identifiers, or web links for publicly available datasets
- A description of any restrictions on data availability
- For clinical datasets or third party data, please ensure that the statement adheres to our policy

doi: 10.17632/dbn4fv37xy.1

# Human research participants

Policy information about studies involving human research participants and Sex and Gender in Research.

| Reporting on sex and gender | N/A |
|---|---|
| Population characteristics | N/A |
| Recruitment | N/A |
| Ethics oversight | N/A |

Note that full information on the approval of the study protocol must also be provided in the manuscript.

# Field-specific reporting

Please select the one below that is the best fit for your research. If you are not sure, read the appropriate sections before making your selection.

☒ Life sciences ☐ Behavioural & social sciences ☐ Ecological, evolutionary & environmental sciences

For a reference copy of the document with all sections, see nature.com/documents/nr-reporting-summary-flat.pdf

# Life sciences study design

All studies must disclose on these points even when the disclosure is negative.

| Sample size | Sample size was determined with a power analysis based off the preliminary data effect size and variability |
|---|---|
| Data exclusions | No data exclusions |
| Replication | Biological replication was achieved by measuring each unique cell, sample, and hippocampal slice once, derived from at least two separate cultures and a minimum of four distinct wells. Sample size was at minimum 3 independent samples. |
| Randomization | Randomization was accomplished by reversing the sample order for every experiment |
| Blinding | Investigators were not blinded to the samples during collection or analysis. Blinding was not performed due to resources constraints combined with the nature of experiments and analysis having low potential for introducing bias |

# Reporting for specific materials, systems and methods

We require information from authors about some types of materials, experimental systems and methods used in many studies. Here, indicate whether each material, system or method listed is relevant to your study. If you are not sure if a list item applies to your research, read the appropriate section before selecting a response.

## Materials & experimental systems

| n/a | Involved in the study |
|---|---|
| ☐ | ☒ Antibodies |
| ☐ | ☒ Eukaryotic cell lines |
| ☒ | ☐ Palaeontology and archaeology |
| ☐ | ☒ Animals and other organisms |
| ☒ | ☐ Clinical data |
| ☒ | ☐ Dual use research of concern |

## Methods

| n/a | Involved in the study |
|---|---|
| ☒ | ☐ ChIP-seq |
| ☒ | ☐ Flow cytometry |
| ☒ | ☐ MRI-based neuroimaging |

# Antibodies

| Antibodies used | CB 2, available at Invitrogen but made in house; Invitrogen cat# 13-730-0.<br>anti-CaMKII (1:2000, BD); BD Transduction Laboratories; #611293; lot# 99135.<br>pT286-CaMKII (1:2500, Phospho-Solutions); #p1005-2886; lot# ks921b.<br>anti-GST (1:2000, Millipore); #AB3282; lot# 3083109<br>pS831-GluA1 (1:2000, Phospho-Solutions); #p1160-831; lot# cs921p.<br>pS1303-GluN2B (1:2000, Millipore); #07-398; lot# 3792158. |
|---|---|

goat-anti mouse (1:10,000 GE); Sheep anti-mouse-HRP; #NA931V; lot# 17205275
goat-anti rabbit (1:10,000 GE); Donkey Anti-Rabbit-HRP; #NA934V; lot# 17469003
CyDye 700 goat anti mouse (1:10,000 Cytiva); #29360784, lot# OF29A-XP
Cydye 800 goat-anti rabbit (1:10,000 Cytiva); #29360790; lot# OF-29C-XP

Validation
Antibodies for CaMKII and pT286 were validated using KO or KI (T286A) samples in ref #51.
anti-GST antibody was validated at Millipore AB3282; RRID: AB_91439
anti-pS831-GluA1 antibody was validated at Phospho-Solutions and by in vitro phosphorylation in ref #51
anti-pS1303-GluN2B (aka NR2B) was validated at Millipore cat. # 07-398 and by in vitro phosphoryation in ref #15

# Eukaryotic cell lines

Policy information about cell lines and Sex and Gender in Research

Cell line source(s)
HEK-293 cells

Authentication
Were not authenticated

Mycoplasma contamination
Were not tested for mycoplasma

Commonly misidentified lines
(See ICLAC register)
No commonly misidentified lines were used in this study (according to ICLAC register)

# Animals and other research organisms

Policy information about studies involving animals; ARRIVE guidelines recommended for reporting animal research, and Sex and Gender in Research

Laboratory animals
Mice; C57BL/6; 8-10 weeks. Wildtype, CaMKII KO, CaMKII T286A, GluN2B KI

Wild animals
N/A

Reporting on sex
Findings of WT and T286A KI animals apply to males. Findings of AAV-injected CaMKII KO apply to both males and females.

Field-collected samples
N/A

Ethics oversight
University of Colorado Institutional Animal Care and Use Committee (IACUC)

Note that full information on the approval of the study protocol must also be provided in the manuscript.

