## [Peer Review File · Nature]

Manuscript Title: LTP induction by structural rather than enzymatic functions of CaMKII

Reviewer Comments & Author Rebuttals

Reviewer Reports on the Initial Version:

Referees' comments:

Referee #1 (Remarks to the Author):

Tullis tested a new idea that CaMKII is involved in the induction of LTP in hippocampal CA1 region solely through its structural activity but not through its kinase activity. The kinase activity is required to phosphorylate T286, which has been shown to render it constitutively active and is required for persistent interaction with NR2B carboxyl tail, but not to phosphorylate other substrates. For this purpose, they combined several new tools. Of note, they used a pharmacological perturbation using a newly developed CaMKII inhibitor, AS283, that blocks the ATP binding to CaMKII. They also used optically activatable PA-CaMKII and NM-PP1/CaMKII F89G.

Overall, I like the story and believe that this paper may make a strong case to Nature. There are, however, issues that needs to be solved before I can totally recommend publication.

1. The authors mention "structural role". But it is not clear what they mean by that. They think NR2B binding is the structural role but it is not clear what the binding does to the structure of the synapse. The same group of scholars proposed in their Nature paper in 2001 that CaMKII binding to NR2B locks CaMKII into an active conformation through T-site interaction. This brings CaMKII near the Ca²⁺-entry site for future activation. But if the kinase activity is not required for what does NR2B binding do? The presented data are very obscure on this point and the authors seem calling anything not explained by the kinase as "structural role". Further characterization on this point is required. Indeed, NR2B KI with mutation that blocks interaction with CaMKII (Halt, 2012), is a complementation to the current study because it does not block kinase activity but does block "structural activity". It still shows partial LTP. It is highly recommended to test this mutant.
2. The stoichiometry between CaMKII and NR2B is very different. At each synapse, there are ~20 NMDA receptor complex. Given both NR2A and NR2B coexist, the number of NR2B molecule per synapse is less. In contrast, there are ~5600 CaMKII (Sheng and Hoogenraad, 2007). Even considering the dodecameric structure of CaMKII, there are ~466 CaMKII holoenzyme. Therefore, there is a large gap between the number of these two molecules and NR2B binding cannot explain synaptic enrichment of CaMKII. Indeed, NR2B KI with mutation that blocks interaction with CaMKII still exhibits partial LTP. There are several other proteins that binds CaMKII in similar fashion such as Tiam1, Densin-180, Rem2 etc. but the authors do not seem to consider about these. Also, recent crystallographic data suggest that there is no distinction between T- and S-site and many of known substrates just interact with a broad binding site in the same fashion as NR2B. This requires more careful clarification.
3. T286 phosphorylation is not required for the initial binding between CaMKII and NR2B but is required for the persistence of the interaction after chelation of Ca²⁺ (Bayer, 2001 and also Hosokawa, 2021). The authors triggered the interaction in HEK293 cells by ionomycin treatment but they did not carry out the washout. They should show the time lapse of change in colocalization before, during the treatment, and washout with normal extracellular solution (or EGTA containing solution) to see the effect of AS105 or AS283. Along the same line, the authors used 100 uM glutamate/10 uM glycine in 1 mM Mg for 45 sec for chemical LTP. This is relatively strong condition. How long the accumulation lasts? The authors need to the time-lapse of accumulation, especially how long it lasts after LTP induction with and without drugs for various mutants.
4. There are several curious results. Why AS283 restores LTP in T286A mutant is not clear.

Especially, there does not seem to be a difference with and without drug up to 10 min. Also, I am not totally convinced by the authors explanation of F89G mutant. Why it rescues LTP see with F89G only? They need a control of rescue with mCherry-WT.

Minor points

1, In Fig. 4, why did the authors not investigate the binding of CaMKII F89G to GluN2B in the presence of NM-PP1 (like in Fig. 2c or 3e). One wonders whether CaMKII F89G can directly bind to GluN2B in the presence of NM-PP1.

2. In addition, the autophosphorylation of CaMKII F89G (pT286) should be validated in Fig. 4b. It should be also investigated whether, in the presence of NM-PP1, CaMKII F89G reduce phosphorylation of CaMKII T286 or its substrates in neuron.

3 It should be kept in mind that AS283 may inhibit not only CaMKII but also other kinase. According to Extended Data Table 1, Flt3, KDR and p70S6K are inhibited over 80% of max by AS283.

4, The data presented in Fig. 4b need to be quantified.

5, What happen if photo-activation of the paCaMKII in neurons from GluN2B KI background (Halt, EMBO J. 2012) or TARPg-8 CaMKII site-dead (Park, Neuron 2016)?

Referee #2 (Remarks to the Author):

The manuscript "LTP is induced by structural rather than enzymatic functions of CaMKII" by Tullis et al. presents data that the catalytic activity of CaMKII is not actually required for the induction of LTP and that the regulated binding of CaMKII to the NR2B subunit is sufficient to induce LTP. This a novel finding that goes against the existing dogma that CaMKII activity and its kinase substrates are critical for LTP expression.

First the authors use a photoactivated CaMKII to induce its binding to the c-tail of GluN2B in a reconstituted system in HEK 293 cells. This effect was inhibited by mutations in CaMKII that inhibit kinase activity or a mutation that does not inhibit kinase activity but inhibits GluN2B binding. Using these same reagents, they show that photoactivation recruits CaMKII to synapses in cultured neurons and also increases spine size, a surrogate for measuring LTP physiologically.

Using a novel CaMKII inhibitor ASN283 that inhibits kinase activity but does not block GluN2B binding to GluN2B in HEK cells and also does not inhibit recruitment of CaMKII recruitment to synapses during chemically induced LTP in neuronal cell cultures.

This inhibitor does not inhibit LTP in hippocampal slices from WT mice but extremely surprisingly it restores LTP in a mutant mouse containing a mutation in the threonine 286 the autophosphorylation site that enhances GluN2B binding.

ASN283 also did not block calcium/calmodulin induced CaMKII binding to GluN2B using purified proteins but actually enhanced binding when compared to ATP.

To further test their hypothesis the authors used a "Shokat" mutant CaMKII that is sensitive to the inhibitor NM-PP1. Remarkably, NM-PP1 does not inhibit ionophore induced CaMKII GluN2B binding or chemical LTP induced recruitment of CaMKII to synapses. Finally, even more surprisingly NM-PP1 rescues LTP in hippocampal slices expressing the Shokat mutant CaMKII suggesting that NM-PP1 did not inhibit but actually enhanced the Shokat CaMKII induction of LTP.

These results are very interesting and intriguing and provide strong evidence suggesting that CaMKII provides a structural role that is essential for LTP expression but I do not think it shows that it is sufficient for LTP expression which is implied in the manuscript. In almost all of these experiments "LTP" is induced by some kind of stimulation that raises intracellular calcium that can be having effects on many other kinases, including other CaMKIIs, PKCs and many other downstream signaling pathways including other calmodulin dependent processes. The sufficiency argument also goes against a very large literature showing that downstream signaling processed including RAS, ERK and PKA signaling is required for structural and physiological LTP induction.

This raises the question of how does CaMKII binding to GluN2B and inducing liquid phase transitions in the PSD structure regulates actin dynamics to change spine size and AMPA receptor recruitment etc. and the other downstream pathways. The authors do not speculate on this or give us a conceptual basis for this idea. It would be great if they could address this in the discussion in the manuscript.

Additional points:

1. ASN283 does inhibit kinase activity for serine S831 and T286 in vitro but the authors do not examine phosphorylation of these sites in neurons and slices under the conditions the examine LTP. They also do not look at the many other CaMKII substrates thought to be important for LTP induction. The inhibitors may not act in vivo as assayed in vitro. This is important to examine, especially for a new relatively uncharacterized inhibitor with amazing properties (enhancing GluN2B binding).

2. The authors refer to old less specific ATP competitive inhibitors such as staurosporine and H7 inhibitors. Based on their work these inhibitors should not block CaMKII binding to GluN2B. However, these inhibitors have been reported to block LTP induction. Can this be discussed.

3. The authors have only examined LTP using field recordings with a single induction protocol. Other induction protocols have been reported to have other signaling requirements. Have the authors looked at other induction protocols or have they used whole cell recording techniques to measure the effect of the inhibitors on LTP induction.

Referee #3 (Remarks to the Author):

This is a timely and detailed analysis of a central pillar of how the field imagines neural plasticity in brain circuits is carried out. Indeed, as the authors introduce, activation and translocation of CaMKII following NMDAR-mediated Ca influx are nearly the only agreed-upon elements of how LTP is induced, and the nearly universal conclusion has been that CaMKII must phosphorylate something in order to enable LTP. Pretty strikingly, there has been little consensus about what those essential targets are, and this decades-long stasis in the field is very nicely explained by the authors' conclusions that in fact it needn't phosphorylate anything after translocation.

As LTP is the premier cellular model of plasticity among only a few compellingly generalizable mechanisms (which might also include LTD and a catch-all basket of homeostatic mechanisms), and is thought to be engaged during innumerable developmental and disease-related events, understanding what CaMKII is doing is extremely important.

I have few complaints or concerns. However, this has high potential to be a landmark paper in the field, and so it is worth bringing up a couple of points to make sure all the bases are covered as befits this future position.

First, one might argue that the authors portray "LTP" as a singular mechanism when in fact many

forms of LTP are induced by various forms of stimuli. The conclusions here are quite stark about toppling current dogma, so it may be appropriate to consider repeating key experiments with another induction protocol. Perhaps those in Fig 3 might be the simplest case. Alternatively, one could imagine that there are edge conditions at the threshold for induction where CaMKII-mediated target phosphorylation is required to augment the structural role, for instance.

Second, the authors convincingly demonstrate that kinase activity of CaMKII is not sufficient or necessary for LTP. However, the mechanism or model beyond that is left more or less entirely unclear (beyond GluN2B binding and the brief reference to another undefined mechanism of liquid-liquid phase separation). At a simple level, reference to only one role strongly suggests the authors believe the LLPS model to be the explanation. Is this the correct interpretation? More broadly, determining the structural mechanism may take great effort and seems well beyond the scope here. However, there is a concern that only insiders in the field will know what to make of this advance, since the implications of overturning the dogma are really not discussed at all. Can the authors explain to a broad audience as well as insiders what it means for us all that this is the case? This is a particular concern given that endogenously, the phosphorylation at T286 is still likely to be critical, if I understand the authors' model correctly. Does the finding here change how we should seek to understand plasticity-related events upstream or downstream of CaMKII translocation? How we pursue therapeutics in this area? The behaviors or conditions in which these mechanisms are engaged? Again, this is not a request for experiments.

Third, the paper is concisely written, very approachable, and authoritative in design and description. It is however, slightly disingenuous to gloss over past debates on this issue, even though it might have been done only for the sake of space. Other authors, perhaps most prominently Yasunori Hayashi, have advanced a structural hypothesis in the past (though I am not aware of suggestions that the kinase activity is entirely unimportant). Related to the above lack of mechanism provided in the paper, the principal suggestion from the Hayashi lab has been the involvement of actin regulation (bundling?) by CaMKIIa; do the authors feel their data supports or should exclude the role of actin or other ideas of how CaMKII mediates non-kinase roles?

Last, there are some minor things to touch up.

- What really is the residual amount of kinase activity in neurons with the treatments and mutations used here? Is no kinase activity really NO kinase activity? How much is necessary to induce LTP? This puts the authors in the position of proving a negative, but the unequivocal statements should be supported as quantitatively as possible in this regard.
- It is slightly unclear in line 232 what the meaning of "information processing" is or why the pT286 reaction is apparently so well suited to it.
- Specificity of AS283. The inhibition curves start in the single-digit nM but then the experiments are at 10 or 30 uM. The table in the supplement shows other enzymes are likely inhibited, potentially even strongly, at these concentrations, though this might not be a fair comparison given the conditions of the kinase activity assay. Some further consideration of whether this impacts the interpretation would be welcomed.
- Fig 2b second lane may be incorrectly labeled. Is it .041 μ M or something else?
- Very trivially, I find the models in Figs 1A and particularly 2A rather difficult to interpret and a bit garish in terms of colors. Perhaps even just making the enzyme itself not colorized would be OK? That doesn't seem to be needed, and precipitates a series of other choices that I think are making the illustration denser are harder to read.

Author Rebuttals to Initial Comments:

Response to review of Tullis et al.,

“LTP induction by structural rather than enzymatic functions of CaMKII”

We were very pleased about the very positive assessment by all reviewers. The remaining points are now addressed as detailed below. This includes addition of new experiments and Figures, now shown in **Figs. 1f,g** and Extended Data (ED) **Fig. 1e** (photo-induced CaMKII movement and spine growth in hippocampal neurons from WT mice vs GluN2B mutant that is incompetent for CaMKII binding);

Fig. 3f (effect of ATP vs AS283 on CaMKII binding to GluN2B S1303A *in vitro*);

ED Figs. 1d,c (photo-induced paCaMKII binding to GluN2B *in vitro*);

ED Fig. 3c (inhibition of pT286 by AS283 in hippocampal neurons);

ED Fig. 3c (inhibition of GluN2B phosphorylation at S1303 by AS283);

ED Fig. 4a (T286 autophosphorylation by WT but not F89G CaMKII and no effects of NM-PP1 on WT);

ED Fig. 4c (rescue of F89G binding to GluN2B *in vitro* by NM-PP1).

Additionally, we included various clarifications in the updated text and changed the title to conform with Nature policy (no active verbs in the title).

Maybe most notable are the additional experiments that further support that GluN2B binding is indeed the structural CaMKII function that is most important (Figs. 1f,g + ED Fig. 1e). This was already suggested by the fact that two CaMKII mutations that disrupt this binding also disrupt light-induced CaMKII movement and spine growth. Now we show the same disruption also by a GluN2B mutation that disrupts the binding on the other end. The main dogma-shifting point of our manuscript remains that enzymatic activity of CaMKII is not required for LTP induction; however, now we also have a better pinpoint on the specific structural function that is required instead.

Response to Referee #1:

We were very pleased that this reviewer did “*like the story*” and did “*believe that this paper may make a strong case to Nature*”. However, before the reviewer “*can totally recommend publication*”, they wanted several issues addressed; this is now done as detailed below.

1. The authors mention “structural role”. But it is not clear what they mean by that. They think NR2B binding is the structural role but it is not clear what the binding does to the structure of the synapse. The same group of scholars proposed in their Nature paper in 2001 that CaMKII binding to NR2B locks CaMKII into an active conformation through T-site interaction. This brings CaMKII near the Ca²⁺-entry site for future activation. But if the kinase activity is not required for what does NR2B binding do? The presented data are very obscure on this point and the authors seem calling anything not explained by the kinase as “structural role”. Further characterization on this point is required. Indeed, NR2B KI with mutation that blocks interaction with CaMKII (Halt, 2012), is a complementation to the current study because it does not block kinase activity but does block “structural activity”. It still shows partial LTP. It is highly recommended to test this mutant.

Response: We have now added important additional experiments in favor of NR2B (=GluN2B) binding as the “structural role” underlying our findings (in new Fig. 1f,g and ED Fig 1e). Specifically, we

tested photoactivation-induced CaMKII movement and spine growth in hippocampal neurons from the suggested NR2B KI mice (termed GluN2B-deltaCaMKII). Notably, both CaMKII movement and spine growth was abolished in the KI mice. This effect appeared to be even stronger than the partially but not completely abolished LTP in slices of these KI mice. This could be explained by compensatory effects in these mice (which could partially circumvent CaMKII requirements after electrical stimulation, but would not be expected to compensate for the response to acute, direct photoactivation of paCaMKII, as this is not an endogenously occurring function and there is thus no pressure for compensation).

While our original results already pointed towards GluN2B binding as the structural role, they fell slightly short of demonstrating it. For this reason, we used somewhat careful wording (i.e. emphasizing “structural functions” as a contrast to “enzymatic functions”, with giving GluN2B binding as one specific potential example for such structural function). We kept this careful wording, but our results now make a much stronger case that the important structural function is indeed GluN2B binding.

2. The stoichiometry between CaMKII and NR2B is very different. At each synapse, there are ~20 NMDA receptor complex. Given both NR2A and NR2B coexist, the number of NR2B molecule per synapse is less. In contrast, there are ~5600 CaMKII (Sheng and Hoogenraad, 2007). Even considering the dodecameric structure of CaMKII, there are ~466 CaMKII holoenzyme. Therefore, there is a large gap between the number of these two molecules and NR2B binding cannot explain synaptic enrichment of CaMKII. Indeed, NR2B KI with mutation that blocks interaction with CaMKII still exhibits partial LTP. There are several other proteins that binds CaMKII in similar fashion such as Tiam1, Densin-180, Rem2 etc. but the authors do not seem to consider about these. Also, recent crystallographic data suggest that there is no distinction between T- and S-site and many of known substrates just interact with a broad binding site in the same fashion as NR2B. This requires more careful clarification.

Response: Indeed, the ratio of CaMKII to GluN2B-containing NMDAR continues to pose somewhat of a conundrum (even though at least in hippocampus, almost all NMDARs are thought to be heterotrimeric and thus contain a GluN2B subunit). Nonetheless, our new data (see point 1) further corroborate previous findings that the CaMKII/GluN2B interaction is necessary for the stimulus-induced CaMKII accumulation at synapses. It is possible that other individual binding reactions contribute to this CaMKII accumulation; however, the only individual binding reaction that has been shown to be necessary is the one with GluN2B, and experiments in heterologous cells suggest that it could also be sufficient.

Additionally, we have now also provided further clarification regarding T- versus S-site distinction (in the first titled subsection of the results), although a more in-depth clarification will likely have to be made elsewhere. Briefly, T286 in the CaMKII regulatory domain binds to the kinase domain in T-site mode in the basal state (holding the regulatory domain in place); however, phosphorylation of T286 upon activation requires binding to the S-site (by definition, as this is the active site mediating substrate phosphorylation). The shorter GluN2B-derived peptide that was used for the crystallography in the work from Margaret Stratton’s laboratory binds preferably in the S-site mode whereas longer GluN2B peptides bind preferably in the T-site mode (according to Bayer et al 2006, J Neurosci). With the brief statement that we have now included we were aiming to strike a balance between sufficiently informing the aficionado while at the same time avoiding unnecessarily distraction for the general reader.

3. T286 phosphorylation is not required for the initial binding between CaMKII and NR2B but is required

for the persistence of the interaction after chelation of Ca²⁺ (Bayer, 2001 and also Hosokawa, 2021). The authors triggered the interaction in HEK293 cells by ionomycin treatment but they did not carry out the washout. They should show the time lapse of change in colocalization before, during the treatment, and washout with normal extracellular solution (or EGTA containing solution) to see the effect of AS105 or AS283. Along the same line, the authors used 100 uM glutamate/10 uM glycine in 1 mM Mg for 45 sec for chemical LTP. This is relatively strong condition. How long the accumulation lasts? The authors need to the time-lapse of accumulation, especially how long it lasts after LTP induction with and without drugs for various mutants.

Response: While these are indeed very interesting questions for detailed follow-up studies, this is well beyond the scope of our current manuscript: As our manuscript is about LTP induction (not maintenance), the duration of the CaMKII accumulation does not further inform our results or our claims. For our current claims, either short or long duration would both be consistent. While previous studies -including our own- would predict a long duration, this does not impact the dogma-shifting claims of our current manuscript regarding CaMKII activity in the induction of LTP one way or the other.

4. There are several curious results. Why AS283 restores LTP in T286A mutant is not clear. Especially, there does not seem to be a difference with and without drug up to 10 min. Also, I am not totally convinced by the authors explanation of F89G mutant. Why it rescues LTP see with F89G only? They need a control of rescue with mCherry-WT.

Response: We have now elaborated more on the mechanism underlying the LTP restoration in T286A mice by AS283 (in new Fig. 3f and ED Fig. 3c). The results lend further support to the notion that the AS283 can enhance GluN2B binding (more than nucleotide) even in absence of pT286. Thereby AS283 can substitute for the structural requirement of pT286 and enable LTP (even though it blocks activity and thereby also the T286 phosphorylation that -in absence of AS283- is required for LTP). This is also supported by our other experiments (in Fig. 6a,b): Addition of AS283 enabled light-induced movement of the photoactivatable paCaMKII T286A mutant (a mutant that did not move in response to light without AS283).

The combination of the F89G mutant and its inhibitor NM-PP1 acts similarly as the combination of T286A and AS283 (but even more exaggerated): F89G is impaired for GluN2B binding (even more strongly than T286A), but this binding is restored by NM-PP1 (as shown in Fig. 4c and new ED Fig. 4c). Importantly, F89G does NOT rescue LTP on its own (see Fig. 5d,e). Instead, addition of NMPP1 to F89G is required to mediate this rescue (see Fig. 5d,e). Importantly, without F89G, NMPP1 does not have this LTP rescuing effect (see Fig. 5b,c). Thus, our current data provide a much better control than mCherry CaMKII WT. (And for practical purposes, adding a control rescue with mCherry-CaMKII WT would require generating CaMKII KO animals, injecting them with virus around week 3, and then doing the recordings after 2 month -in addition to first actually generating and producing the virus-, thus making this a very extensive and time-consuming addition.)

Minor points

1, In Fig. 4, why did the authors not investigate the binding of CaMKII F89G to GluN2B in the presence of NM-PP1 (like in Fig. 2c or 3e). One wonders whether CaMKII F89G can directly bind to GluN2B in the presence of NM-PP1.

Response: Now added as requested (in new ED Fig. 4c). We had previously shown only the

binding studies in HEK cell; now a biochemical *in vitro* binding study is added.

2. *In addition, the autophosphorylation of CaMKII F89G (pT286) should be validated in Fig. 4b. It should be also investigated whether, in the presence of NM-PP1, CaMKII F89G reduce phosphorylation of CaMKII T286 or its substrates in neuron.*

Response: Fig. 4b showed phosphorylation of GluA1 S831. A similar experiment showing the effect on pT286 is now added as requested (in new ED Fig. 4a). While F89G showed some apparent minimal phosphorylation of GluA1 S831, no significant T286 phosphorylation was detected at all.

3 *It should be kept in mind that AS283 may inhibit not only CaMKII but also other kinase. According to Extended Data Table 1, Flt3, KDR and p70S6K are inhibited over 80% of max by AS283.*

Response: The updated legend to Extended Data Table 1 now clarifies that Flt3 (the one AS105-inhibited kinase that is still inhibited by AS283) is not significantly expressed in the hippocampus. Most importantly, however, we show that AS283 does NOT inhibit LTP and inhibition of other kinases would only pose a potential problem if we instead had seen an effect on LTP; as is, our data indicate that enzymatic activity of neither CaMKII nor Flt3 (nor of any other kinases that may be inhibited by AS283) is required.

4, *The data presented in Fig. 4b need to be quantified.*

Response: The quantification of the GluA1 S831 phosphorylation is now shown in new in new ED Fig. 4a, together with the added quantification of pT286. While F89G showed some apparent minimal phosphorylation of GluA1 S831, no significant T286 phosphorylation was detected at all.

5, *What happen if photo-activation of the paCaMKII in neurons from GluN2B KI background (Halt, EMBO J. 2012) or TARPg-8 CaMKII site-dead (Park, Neuron 2016)?*

Response: We have now done this for the GluN2B KI mice (see also point 1 and 2). Similar as introducing CaMKII mutants that impair GluN2B binding, both light-induced CaMKII movement and spine growth is blocked in neurons with this GluN2B mutation that inhibits CaMKII binding. We do not have access to TARPg-8 CaMKII site-dead mice; however, TARPg-8 phosphorylation is included in our discussion.

Response to Referee #2:

We were very pleased that this reviewer found that our *“results are very interesting and intriguing”* and *“provide strong evidence suggesting that CaMKII provides a structural role that is essential for LTP expression”*; and that *“this is a novel finding that goes against the existing dogma”*. Remaining points are now addressed as detailed below.

0. *These results are very interesting and intriguing and provide strong evidence suggesting that CaMKII provides a structural role that is essential for LTP expression but I do not think it shows that it is sufficient for LTP expression which is implied in the manuscript. In almost all of these experiments “LTP” is induced by some kind of stimulation that raises intracellular calcium that can be having effects on many other*

kinases, including other CaMKIIs, PKCs and many other downstream signaling pathways including other calmodulin dependent processes. The sufficiency argument also goes against a very large literature showing that downstream signaling processed including RAS, ERK and PKA signaling is required for structural and physiological LTP induction.

This raises the question of how does CaMKII binding to GluN2B and inducing liquid phase transitions in the PSD structure regulates actin dynamics to change spine size and AMPA receptor recruitment etc. and the other downstream pathways. The authors do not speculate on this or give us a conceptual basis for this idea. It would be great if they could address this in the discussion in the manuscript.

Response: Indeed, our main point is the necessity, and our evidence for this part is much stronger. However, we believe that the issue with “sufficiency” is mainly semantics, and our revised manuscript clarifies this distinction by more careful wording: Of course structural CaMKII functions are not completely sufficient for LTP (as LTP obviously requires at least also AMPA receptors for its expression, in addition to various degrees of the other mentioned signaling molecules). However, among the CaMKII functions that are required for LTP, the structural ones are sufficient as the enzymatic CaMKII functions are not necessary (even though CaMKII is still required). Accordingly, this is now worded more accurately in the first sentence of the discussion (and in the last sub-section of the results).

We have now added some discussion of the actin in context of the possible downstream functions, but the details of the structural mechanisms remain to be elucidated (see also response to reviewer 3 points 2 and 3). While LLPS provides an especially intriguing possibility (which is specifically mentioned at the end of our discussion), we do not believe that further extended speculation would be beneficial to the reader here.

Additional points:

1. ASN283 does inhibit kinase activity for serine S831 and T286 in vitro but the authors do not examine phosphorylation of these sites in neurons and slices under the conditions they examine LTP. They also do not look at the many other CaMKII substrates thought to be important for LTP induction. The inhibitors may not act in vivo as assayed in vitro. This is important to examine, especially for a new relatively uncharacterized inhibitor with amazing properties (enhancing GluN2B binding).

Response: We have now performed additional experiments that show inhibition of pT286 by AS283 in hippocampal neurons (see new ED Fig. 3c). The observed inhibition was similar to the inhibition of CaMKII activity seen in our *in vitro* experiments. Notably, the effectiveness of AS283 also in hippocampal slices is indicated by the fact that AS283 actually rescues LTP in slices from T286A mutant mice. These experiments also show that the lack of LTP inhibition in wildtype mice is not due to potential residual T286 phosphorylation.

2. The authors refer to old less specific ATP competitive inhibitors such as staurosporine and H7 inhibitors. Based on their work these inhibitors should not block CaMKII binding to GluN2B. However, these inhibitors have been reported to block LTP induction. Can this be discussed.

Response: Indeed, one of the original papers that used a CaMKII inhibitor to block LTP also showed that H7 also blocks LTP (Malinow et al 1989). This is now specifically mentioned and cited (in the results section, right before the description of no block of LTP with AS283). One possible explanation is

that LTP requires the activity of one or more of the many other kinases that are inhibited by H7 (a possibility that is not disputed by the claims of our manuscript).

3. The authors have only examined LTP using field recordings with a single induction protocol. Other induction protocols have been reported to have other signaling requirements. Have the authors looked at other induction protocols or have they used whole cell recording techniques to measure the effect of the inhibitors on LTP induction.

Response: In addition to assessing LTP by field recordings in slices, we have imaged structural LTP in cultured neurons in response to light stimulation of pCaMKII. Thus, we have examined two very distinct aspects and induction protocols of LTP. Notably, compared to whole cell recordings, both methods cause much less disturbance of the intracellular signaling environment.

Response to Referee #3:

We were extremely pleased that this reviewer found that “*understanding what CaMKII is doing is extremely important*” and that our study “*has high potential to be a landmark paper*” that nicely resolves “*decades-long stasis in the field*”. Remaining points “*to make sure all bases are covered*” are now addressed as detailed below.

First, one might argue that the authors portray “LTP” as a singular mechanism when in fact many forms of LTP are induced by various forms of stimuli. The conclusions here are quite stark about toppling current dogma, so it may be appropriate to consider repeating key experiments with another induction protocol. Perhaps those in Fig 3 might be the simplest case. Alternatively, one could imagine that there are edge conditions at the threshold for induction where CaMKII-mediated target phosphorylation is required to augment the structural role, for instance.

Response: In order to clarify this issue better, we provide additional discussion of our stimulation LTP protocols (after the first sentence of the discussion). Notably, our stimuli (2x HFS in slices; light stimulation in cultures) are rather mild “middle of the road” conditions (and not an extreme protocol that can drive LTP even with tail-less AMPA receptors). As such, the chosen LTP induction conditions are sufficient to show that the currently prevailing dogma is incorrect. Searching for possible threshold conditions (in which both structural and enzymatic CaMKII functions are required) could be interesting for future studies but appears beyond the scope of the current manuscript.

Second, the authors convincingly demonstrate that kinase activity of CaMKII is not sufficient or necessary for LTP. However, the mechanism or model beyond that is left more or less entirely unclear (beyond GluN2B binding and the brief reference to another undefined mechanism of liquid-liquid phase separation). At a simple level, reference to only one role strongly suggests the authors believe the LLPS model to be the explanation. Is this the correct interpretation? More broadly, determining the structural mechanism may take great effort and seems well beyond the scope here. However, there is a concern that only insiders in the field will know what to make of this advance, since the implications of overturning the dogma are really not discussed at all. Can the authors explain to a broad audience as well as insiders what it means for us all that this is the case? This is a particular concern given that

endogenously, the phosphorylation at T286 is still likely to be critical, if I understand the authors' model correctly. Does the finding here change how we should seek to understand plasticity-related events upstream or downstream of CaMKII translocation? How we pursue therapeutics in this area? The behaviors or conditions in which these mechanisms are engaged? Again, this is not a request for experiments.

Response: Indeed, the exact downstream mechanisms of the CaMKII structural functions that are required for LTP remain unclear (and are beyond the scope of this study, as indicated by the reviewer). Our dogma-toppling contribution is that enzymatic CaMKII activity is not required. This helps explain why 35 years of extensive research on identifying the relevant enzymatic CaMKII substrate(s) in LTP has come up rather empty-handed (with the 2 most noteworthy substrates discussed in our discussion). This hopefully will help re-direct the efforts to identify the structural mechanisms implicated here. LLPS is one such candidate mechanism, and it has received much recent attention as a novel organizing principle at the synapse. As such, LLPS is certainly especially intriguing. However, just because we can (currently) not come up with any other better specific mechanism underlying the structural CaMKII functions in LTP does not mean that LLPS is indeed the only one. With the current brevity of the description of LLPS we wish to reflect both the intrigue of this particular potential mechanism and the fact that we do not regard the actual mechanism as settled.

Third, the paper is concisely written, very approachable, and authoritative in design and description. It is however, slightly disingenuous to gloss over past debates on this issue, even though it might have been done only for the sake of space. Other authors, perhaps most prominently Yasunori Hayashi, have advanced a structural hypothesis in the past (though I am not aware of suggestions that the kinase activity is entirely unimportant). Related to the above lack of mechanism provided in the paper, the principal suggestion from the Hayashi lab has been the involvement of actin regulation (bundling?) by CaMKIIa; do the authors feel their data supports or should exclude the role of actin or other ideas of how CaMKII mediates non-kinase roles?

Response: We have now added discussion of F-actin functions (with references to work from the Hayashi lab) near the end of our discussion. Briefly, any structural LTP mechanism must ultimately affect F-actin, however, the F-actin binding/bundling by CaMKII is likely not involved in the mechanisms studied here (as the F-actin binding is mainly mediated by a different isoform and is regulated oppositely from the GluN2B binding that is instead implicated by our results).

Last, there are some minor things to touch up.

-a) What really is the residual amount of kinase activity in neurons with the treatments and mutations used here? Is no kinase activity really NO kinase activity? How much is necessary to induce LTP? This puts the authors in the position of proving a negative, but the unequivocal statements should be supported as quantitatively as possible in this regard.

Response: Our results suggest that no kinase activity indeed means NO kinase activity. (At the very least, they suggest that IF there was any residual CaMKII that would be required, it would be undetectable). Among the better arguments for this is the F89G mutant combination with the NMPP-1 inhibitor: the LTP mechanism for this mutant (that is already only minimally enzymatically active) can be rescued by FURTHER inhibiting it with NM-PP1. At the same time, this rescue shows that NM-PP1 effectively engages the target. A similar argument can be made for the combination of the T286A

mutant with the AS283 inhibitor, albeit not as strongly as for the F89G/NMPP-1 combination. Further arguments in favor of this can be found in the response to minor point c below.

-b) It is slightly unclear in line 232 what the meaning of “information processing” is or why the pT286 reaction is apparently so well suited to it.

Response: For clarification, we have now changed “information processing” to “signal processing” and contrasted it with “LTP maintenance”. Additionally, we now briefly mention the biochemical features of the pT286 reaction that makes it well suited for signal processing. The three references cited each provide in-depth discussion of the statement (in case the statement sparks further interest in the reader).

-c) Specificity of AS283. The inhibition curves start in the single-digit nM but then the experiments are at 10 or 30 μ M. The table in the supplement shows other enzymes are likely inhibited, potentially even strongly, at these concentrations, though this might not be a fair comparison given the conditions of the kinase activity assay. Some further consideration of whether this impacts the interpretation would be welcomed.

Response: Pharmacologically sound doses for near-complete block of activity are typically considered to be 50-1,000fold of the K_i or IC_{50} . Importantly, even at 30 μ M, AS283 did NOT block LTP induction in WT slices. Using a high concentration is a problem only when there is an effect and there is a question about selectivity. This is not the case here. On the contrary, here, our results indicate that completely blocking CaMKII activity at these high concentrations still allows induction of LTP (in direct support of our main conclusion). Nonetheless, we further clarified the potential effects on other enzymes in the legend of our supplementary table: Selectivity of AS283 is further increased compared to AS105; and the one remaining strongly cross-inhibited kinase (Flt3) is not significantly expressed in hippocampus.

-d) Fig 2b second lane may be incorrectly labeled. Is it .041 μ M or something else?

Response: Indeed, this should have been .041 μ M (not .41 μ M). Now corrected accordingly (to 0.04 μ M, i.e. rounded to two digits, in order to avoid implying more accuracy than is warranted).

-e) Very trivially, I find the models in Figs 1A and particularly 2A rather difficult to interpret and a bit garish in terms of colors. Perhaps even just making the enzyme itself not colored would be OK? That doesn't seem to be needed, and precipitates a series of other choices that I think are making the illustration denser are harder to read.

Response: The enzyme is now shown in grey (instead of colored) as suggested.

Reviewer Reports on the First Revision:

Referees' comments:

Referee #1 (Remarks to the Author):

The author addressed issues I raised. Although I still do not fully agree with their view and feel there are some awkward points, results are results. This paper will certainly provoke discussion and criticism, but that is the role of a high-profile journal like Nature. I propose to publish this paper.

Referee #2 (Remarks to the Author):

The authors have responded to many of my comments and have improved the manuscript. However, I come away a little unsatisfied. Some of their arguments are weak and some responses inadequately address my comments and similar comments from the other reviewers.

The authors argue they look at several forms of LTP induction but light stimulation and chemical LTP are artificial modes of LTP induction and are not the classical forms of LTP induction. Whole-cell recording of LTP induction has been used for decades as a major model of LTP induction but is not used here.

In response to reviewer #1's comment, "Indeed, NR2B KI with a mutation that blocks interaction with CaMKII (Halt, 2012), is a complementation to the current study because it does not block kinase activity but does block "structural activity". It still shows partial LTP. It is highly recommended to test this mutant."

They respond "This could be explained by compensatory effects in these mice (which could partially circumvent CaMKII requirements after electrical stimulation, but would not be expected to compensate for the response to acute, direct photoactivation of paCaMKII, as this is not an endogenously occurring function and there is thus no pressure for compensation)." This is rather a weak hand-waving response.

They claim that no other CaMKII substrate is required for LTP induction. This is a very bold claim as they have not examined even a few well-known LTP induction-dependent CaMKII substrates during LTP. They have only looked at GluA1 S831, and even there, they see residual phosphorylation. What other substrates have residual or even higher phosphorylation? What about TARPs and other characterized sites that may be important? As one other reviewer stated, "How much phosphorylation is enough?" I don't want them to do phosphopeptide proteomics but looking in slices at some of these CaMKII phosphorylation sites is a critical experiment.

Finally, how these structural changes regulate potentiation is still a total black box. There is not even a framework or a hypothesis of how this occurs. This makes me uneasy.

Referee #3 (Remarks to the Author):

This was a responsive review, and the authors have addressed my concerns.

Author Rebuttals to First Revision:

Response to review of Tullis et al.,

“LTP induction by structural rather than enzymatic functions of CaMKII”

We were very pleased about the continued interest in our manuscript (and with the fact that two of the referees support publication of our manuscript in the present form). Like the referees, we are big subscribers to the concept of "extraordinary claims require extraordinary evidence". And we actually provide such evidence in this manuscript, as briefly recapitulated here:

We show that a compound that inhibits CaMKII enzymatic activity but which enhances GluN2B binding can **restore** LTP in the CaMKII T286A mutant mice! This provides unequivocal evidence for LTP induction by structural rather than enzymatic CaMKII functions. And this is just one of four of our examples for ATP-competitive CaMKII inhibitors restoring LTP mechanisms for CaMKII mutants that otherwise do not support LTP. Thus, our cumulative evidence goes way beyond “just” (i) showing that ATP-competitive CaMKII inhibitors (which do not interfere with GluN2B binding) do not block LTP and (ii) delineating that all other prior LTP-inhibiting interference with CaMKII activity also interfered with GluN2B binding (and thus, that there is actually no evidence for the widely held but erroneous belief that enzymatic activity of CaMKII would be required for LTP).

Response to Referees #1 and #3:

We were very pleased that both referees now support publication of our paper, and that both referees found that we fully addressed their remaining issues. We appreciate the specific statements by referee #1 that *“the authors addressed issues I raised”* and that they *“propose to publish this paper”*; and by referee #3 that *“this was a responsive review, and the authors have addressed my concerns”*.

Response to Referee #2:

We were very pleased that this referee originally found that our *“results are very interesting and intriguing”* and *“provide strong evidence suggesting that CaMKII provides a structural role that is essential for LTP expression”*; and that *“this is a novel finding that goes against the existing dogma”*.

For our revision, the referee also acknowledged that *“the authors have responded to many of my comments and have improved the manuscript”*, however, unfortunately, this referee still felt to *“come away a little unsatisfied”*. In our detailed response below, we are further addressing these remaining issues more clearly. However, we wish to point out that three of the four remaining points were originally raised by the other referees, who were actually satisfied already by our original detailed response.

1.) *The authors argue they look at several forms of LTP induction but light stimulation and chemical LTP are artificial modes of LTP induction and are not the classical forms of LTP induction. Whole-cell recording of LTP induction has been used for decades as a major model of LTP induction but is not used here.*

Response: Indeed, whole-cell recordings of LTP induction has been used for decades. However, in terms of being the most “classical form of LTP induction”, the extracellular field recordings that we used here still win by a mile. This does not mean that whole-cell recordings don’t have their place (or don’t have certain advantages for certain questions). However, field recordings are arguably more physiological, because they do not disrupt the endogenous intracellular milieu. Additionally, it is entirely unclear what advantages whole-cell recordings would provide here or what they could add to our current conclusions.

In this context, we do not make any formal claim that all forms of LTP are dependent on structural rather than enzymatic CaMKII functions (and this is also reflected in the new title that we changed after the initial review). Our findings show, surprisingly but unequivocally, that we CAN induce LTP with structural rather than enzymatic CaMKII functions. Thus, even if future studies found a specific form of LTP that does require enzymatic CaMKII activity, such finding would not negate our current claims. For this same reason, whole-cell recordings cannot provide a meaningful test of our current claims.

Regarding chemical LTP and optogenetic stimulation with light: These are indeed more artificial forms of LTP (although unlike whole-cell recordings, they also do not disrupt the endogenous intracellular milieu), but as such they provide further evidence for the general applicability of our mechanistic findings. Most importantly, these additional lines of evidence do not stand alone, but are backed up by classical field recordings in acute hippocampal slices.

2.) In response to reviewer #1's comment, "Indeed, NR2B KI with a mutation that blocks interaction with CaMKII (Halt, 2012), is a complementation to the current study because it does not block kinase activity but does block "structural activity". It still shows partial LTP. It is highly recommended to test this mutant." They respond "This could be explained by compensatory effects in these mice (which could partially circumvent CaMKII requirements after electrical stimulation, but would not be expected to compensate for the response to acute, direct photoactivation of paCaMKII, as this is not an endogenously occurring function and there is thus no pressure for compensation)." This is rather a weak hand-waving response.

Response: First, we wish to point out that reviewer #1 (who made this comment originally) was actually satisfied with our response. Second, in response to the original point, we did not only provide the here-critiqued detailed discussion, but also the actually requested experiments (using the requested NR2B/GluN2B KI mutant). Our results with the paCaMKII in neurons from this mutant fully support our conclusion, and the "weak" discussion is only made in the response to the reviewer, not in the paper. What we were pointing out there is that our results provide even stronger evidence for involvement of GluN2B binding than the paper that originally described the impaired LTP in slices from these KI mice (ref#22: Halt et al., 2012). Evoking partial compensation may appear hand-waving, but it is a common effect. In this case, partial compensation in the constitutive KI mice is directly supported by the fact that more acute ways to interfere with CaMKII/GluN2B binding indeed interfered with LTP even more strongly (ref#23: Incontro et al., 2018; ref#21: Barria et al., 2005; and also Sanhueza et al 2012, J Neurosci 31:9179; as well as Barcomb et al 2016, JBC 291:16082), similar to what was seen in our own new findings that we added in response to the initial review.

3.) They claim that no other CaMKII substrate is required for LTP induction. This is a very bold claim as they have not examined even a few well-known LTP induction-dependent CaMKII substrates during LTP. They have only looked at GluA1 S831, and even there, they see residual phosphorylation. What other substrates have residual or even higher phosphorylation? What about TARPs and other characterized sites that may be important? As one other reviewer stated, "How much phosphorylation is enough?" I don't want them to do phosphopeptide proteomics but looking in slices at some of these CaMKII phosphorylation sites is a critical experiment.

Response: Again, this point was raised as a "minor issue" by reviewer #3. We have responded to it in much detail, and reviewer #3 was highly satisfied by our response (specifically stating that that "this was a responsive review, and the authors have addressed my concerns"). {REDACTED} Here, three brief new points for our updated additional response:

(i) We already showed that our inhibitor effectively inhibits phosphorylation reactions by CaMKII (even though we cannot specifically test this for TARP-gamma8, due to lack of good commercially available antibodies).

(ii) It is essential to demonstrate successful inhibition if an inhibitor does NOT have a functional effect. Here, we show that two different inhibitors actually rescue LTP for CaMKII mutants that otherwise do not support LTP (and we show this in 3 different experimental systems). Thus, the effectiveness of our inhibitor is demonstrated also functionally.

(iii) As we will discuss more clearly in a revised manuscript, while the claims of a TARP-gamma8 paper in Neuron {REDACTED} (citation #37) are in conflict with our results {REDACTED} and with the results of citation #38, their actual results are NOT in conflict with our findings at all.

The details to point (iii) go as follows. Citation #37 shows that chemical LTP stimulation increases TARP-gamma8 phosphorylation and that this is inhibited by KN93. However, KN93 inhibits not only CaMKII but also PKC (Brooks and Tavalin, JBC 2011), another kinase known to phosphorylate TARP-gamma8 at the same residues. Notably, no other inhibitor (or a control substance) was used. Most importantly, the cLTP-induced increase appeared rather modest, with ~75% of the maximal phosphorylation already seen under basal conditions, and this basal phosphorylation was NOT sensitive to KN93. Thus, the demonstration that mutation of the phosphorylation sites reduced LTP by half suggests that the phosphorylation contributes to LTP, but it shows neither that LTP requires phosphorylation by CaMKII nor the increase in phosphorylation during LTP.

In further addition to point (iii), as LTP is “only” reduced by 50%, this would indicate that TARP phosphorylation could contribute to LTP but that it is not required (which would be consistent with our claims). One could argue that this is due to a compensation mechanism, which is indeed common. However, in this case (and in contrast to manipulations of CaMKII/GluN2B binding), acute manipulations that would circumvent compensation had no effect on LTP at all (citation #38). This speaks not only against compensation, but questions any contribution to LTP. Interestingly (and consistent with the high level of basal TARP phosphorylation found in citation #37), citation #38 instead found a reduction in basal synaptic transmission. Thus, in summary, while the situation with TARP phosphorylation may be less clear than with phosphorylation of another prominent CaMKII substrate, GluA1 S831 (which has been demonstrated to be not required for LTP), there is no conclusive evidence for acute contribution to LTP and even the results from citation #37 are consistent with our current findings (even if their claims that they made in the title of the manuscript may not be). {REDACTED} Our improved discussion describes these issues more clearly.

For further details to point (ii), we not only show that AS283 (which inhibits CaMKII but also enhance its GluN2B binding) does not inhibit LTP in wildtype hippocampus; we additionally show that it restores LTP in hippocampus from CaMKII T286A mutant mice (which otherwise do not show LTP). This experiment alone is sufficient to fundamentally substantiate our claims. But we go further: We show that re-expression of a “Shokat mutant” of CaMKII (which impairs both CaMKII enzymatic activity and GluN2B binding) in CaMKII KO mice rescued LTP, but only in presence of a “Shokat inhibitor” that blocks activity further while enabling GluN2B binding. Additionally, we show corresponding pharmacogenetic experiments with photoactivatable CaMKII in cultures. Thus, the effectiveness of our inhibitors in slices and in neurons is well established.

For further details to point (i), while direct probing of TARP-gamma8 phosphorylation is not technically feasible due to lack of commercially available antibodies, this should not be necessary (see the points above). However, even if it was possible, the results would not significantly further test our claims, as either result would be consistent: Even if our inhibitor does not reduce TARP-gamma8 phosphorylation, this would NOT indicate that our inhibitor is not effective (see point ii above) but that the phosphorylation is mediated by another kinase (for instance by PKC, which can phosphorylate the same sites and which is inhibited by the KN93 inhibitor of CaMKII that was used to implicate CaMKII; see

point iii). Thus, the proposed experiments on TARP phosphorylation may be interesting, but are for a different story, as they do not affect our current conclusions one way or the other.

4.) Finally, how these structural changes regulate potentiation is still a total black box. There is not even a framework or a hypothesis of how this occurs. This makes me uneasy.

Response: Our findings suggest that the focus can now shift from a search of a key substrate of enzymatic activity of CaMKII (other than itself) to the question of how the CaMKII-GluN2B structure effects synaptic changes that lead to increase synaptic AMPA receptors. And we actually point towards a framework for the structural changes (at the end of our discussion section): the co-condensation of CaMKII with GluN2B (and likely other synaptic proteins) via liquid-liquid phase separation that was recently described in several prominent publications. We do not elaborate on the specific model, because this is detailed in the cited publications; and because our contribution is not toward this specific structural model itself; instead we are showing that enzymatic CaMKII activity is not required at all (thereby elevating this new structural model, even though at this point, this highly intriguing specific new structural model also still remains rather speculative, in our opinion).

Again, this point is also related to a point originally made by another referee (referee #3, in their second original point), and this referee was satisfied by our response. While we understand the “unease” regarding our dogma-shifting findings, similar unease should have been generated by the fact that there are not really any clear CaMKII phosphorylation sites that are required for LTP (despite over four decades of searching; with the single most prominent LTP-related site, GluA1 S831, was eventually found not to be required for LTP). The only exception is the T286 autophosphorylation site, but -as shown here- this was required only for the structural function, not for extending enzymatic activity. Moreover, we provide a clear delineation that any previously described interference with enzymatic CaMKII activity that inhibited LTP also inhibited GluN2B binding. As such, even without our manuscript, there really is no prior evidence at all for the widely held but erroneous belief that enzymatic CaMKII activity is required for LTP. This was the reason why we set out to distinguish between the roles of enzymatic activity and GluN2B binding in the first place. While we were similarly flabbergasted by our results, we now support our initial unexpected finding (of no LTP inhibition by CaMKII inhibitors that do not interfere with GluN2B binding) by four independent sets of rescue experiments that clearly demonstrate that GluN2B binding is essential for LTP mechanisms whereas enzymatic activity is not.

{REDACTED}